# Cytoplasmic mRNA decay represses RNA polymerase II transcription during early apoptosis

Christopher Duncan-Lewis[1], Ella Hartenian[1], Valeria King[1], Britt A Glaunsinger[1,2,3]*

[1]Department of Molecular and Cell Biology; University of California, Berkeley, Berkeley, United States; [2]Department of Plant and Microbial Biology; University of California, Berkeley, Berkeley, United States; [3]Howard Hughes Medical Institute, Berkeley, Berkeley, United States

**Abstract** RNA abundance is generally sensitive to perturbations in decay and synthesis rates, but crosstalk between RNA polymerase II transcription and cytoplasmic mRNA degradation often leads to compensatory changes in gene expression. Here, we reveal that widespread mRNA decay during early apoptosis represses RNAPII transcription, indicative of positive (rather than compensatory) feedback. This repression requires active cytoplasmic mRNA degradation, which leads to impaired recruitment of components of the transcription preinitiation complex to promoter DNA. Importin α/β-mediated nuclear import is critical for this feedback signaling, suggesting that proteins translocating between the cytoplasm and nucleus connect mRNA decay to transcription. We also show that an analogous pathway activated by viral nucleases similarly depends on nuclear protein import. Collectively, these data demonstrate that accelerated mRNA decay leads to the repression of mRNA transcription, thereby amplifying the shutdown of gene expression. This highlights a conserved gene regulatory mechanism by which cells respond to threats.

*For correspondence: glaunsinger@berkeley.edu

**Competing interests:** The authors declare that no competing interests exist.

## Introduction

Gene expression is often depicted as a unidirectional flow of discrete stages: DNA is first transcribed by RNA polymerase II (RNAPII) into messenger RNA (mRNA), which is processed and exported to the cytoplasm where it is translated and then degraded. However, there is a growing body of work that reveals complex cross talk between the seemingly distal steps of mRNA transcription and decay. For example, the yeast Ccr4-Not deadenylase complex, which instigates basal mRNA decay by removing the poly(A) tails of mRNAs (*Tucker et al., 2001*), was originally characterized as a transcriptional regulator (*Collart and Stmhp, 1994*; *Denis, 1984*). Other components of transcription such as RNAPII subunits and gene promoter elements have been linked to cytoplasmic mRNA decay (*Bregman et al., 2011*; *Lotan et al., 2005*; *Lotan et al., 2007*), while the activity of cytoplasmic mRNA degradation machinery such as the cytoplasmic 5′−3′ RNA exonuclease Xrn1 can influence the transcriptional response (*Haimovich et al., 2013*; *Sun et al., 2012*).

The above findings collectively support a model in which cells engage a buffering response to reduce transcription when mRNA decay is slowed, or reduce mRNA decay when transcription is slowed to preserve the steady state mRNA pool (*Haimovich et al., 2013*; *Hartenian and Glaunsinger, 2019*). While much of this research has been performed in yeast, the buffering model is also supported by studies in mouse and human cells (*Helenius et al., 2011*; *Singh et al., 2019*). In addition to bulk changes to the mRNA pool, compensatory responses can also occur at the individual gene level to buffer against aberrant transcript degradation. Termed 'nonsense-induced

transcription compensation' (NITC; *Wilkinson, 2019*), this occurs when nonsense-mediated mRNA decay leads to transcriptional upregulation of genes with some sequence homology to the aberrant transcript (*El-Brolosy et al., 2019*; *Ma et al., 2019*).

A theme that unites much of the research linking mRNA decay to transcription is homeostasis; perturbations in mRNA stability are met with an opposite transcriptional response in order to maintain stable mRNA transcript levels. However, there are cellular contexts in which homeostasis is not beneficial, for example during viral infection. Many viruses induce widespread host mRNA decay (*Narayanan and Makino, 2013*) and co-opt the host transcriptional machinery (*Harwig et al., 2017*) in order to express viral genes. Indeed, infection with mRNA decay-inducing herpesviruses or expression of broad-acting viral ribonucleases in mammalian cells causes RNAPII transcriptional repression in a manner linked to accelerated mRNA decay (*Abernathy et al., 2015*; *Hartenian et al., 2020*). It is possible that this type of positive feedback represents a protective cellular shutdown response, perhaps akin to the translational shutdown mechanisms that occur upon pathogen sensing (*Walsh et al., 2013*). A central question, however, is whether transcriptional inhibition upon mRNA decay is restricted to infection contexts, or whether it is also engaged upon other types of stimuli.

The best-defined stimulus known to broadly accelerate cytoplasmic mRNA decay outside of viral infection is induction of apoptosis. Overall levels of poly(A) RNA are reduced rapidly after the induction of extrinsic apoptosis via accelerated degradation from the 3' ends of transcripts (*Thomas et al., 2015*). The onset of accelerated mRNA decay occurs coincidentally with mitochondrial outer membrane depolarization (MOMP) and requires release of the mitochondrial exonuclease polyribonucleotide nucleotidyltransferase 1 (PNPT1) into the cytoplasm. PNPT1 then coordinates with other 3' end decay machinery such as DIS3L2 and terminal uridylyltransferases (TUTases; *Liu et al., 2018*; *Thomas et al., 2015*). Notably, mRNA decay occurs before other hallmarks of apoptosis including phosphatidylserine (PS) externalization and DNA fragmentation, but likely potentiates apoptosis by reducing the expression of unstable anti-apoptotic proteins such as MCL1 (*Thomas et al., 2015*).

Here, we used early apoptosis as a model to study the impact of accelerated cytoplasmic mRNA decay on transcription. We reveal that under conditions of increased mRNA decay, there is a coincident decrease in RNAPII transcription, indicative of positive feedback between mRNA synthesis and degradation. Using decay factor depletion experiments, we demonstrate that mRNA decay is required for the transcriptional decrease and further show that transcriptional repression is associated with reduced RNAPII polymerase occupancy on promoters. This phenotype requires ongoing nuclear-cytoplasmic protein transport, suggesting that protein trafficking may provide the signal linking cytoplasmic decay to transcription. Collectively, our findings elucidate a distinct gene regulatory mechanism by which cells sense and respond to threats.

## Results

### mRNA decay during early apoptosis is accompanied by reduced synthesis of RNAPII transcripts

To induce widespread cytoplasmic mRNA decay, we initiated rapid extrinsic apoptosis in HCT116 colon carcinoma cells by treating them with TNF-related apoptosis inducing ligand (TRAIL). TRAIL treatment causes a well-characterized progression of apoptotic events including caspase cleavage and mitochondrial outer membrane permeabilization or 'MOMP' (*Albeck et al., 2008*; *Thomas et al., 2015*). It is MOMP that stimulates mRNA decay in response to an apoptosis-inducing ligand (*Figure 1A*), partly by releasing the mitochondrial 3'–5' RNA exonuclease PNPT1 into the cytoplasm (*Liu et al., 2018*; *Thomas et al., 2015*). A time-course experiment in which cells were treated with 100 ng/mL TRAIL for increasing 30 min increments showed activation of caspase 8 (CASP8) and caspase 3 (CASP3) by 1.5 hr (*Figure 1B*), as measured by disappearance of the full-length zymogen upon cleavage (*Kim et al., 2000*; *Thomas et al., 2015*). In agreement with *Liu et al., 2018*, RT-qPCR performed on total RNA revealed a coincident decrease in the mRNA levels of several housekeeping genes (*ACTB, GAPDH, EEF1A, PPIA, CHMP2A, DDX6, RPB2,* and *RPLP0*) beginning 1.5 hr after TRAIL was applied (*Figure 1C*, *Figure 1—figure supplement 1A*). Fold changes were calculated in reference to 18S ribosomal RNA (rRNA), which has been shown to

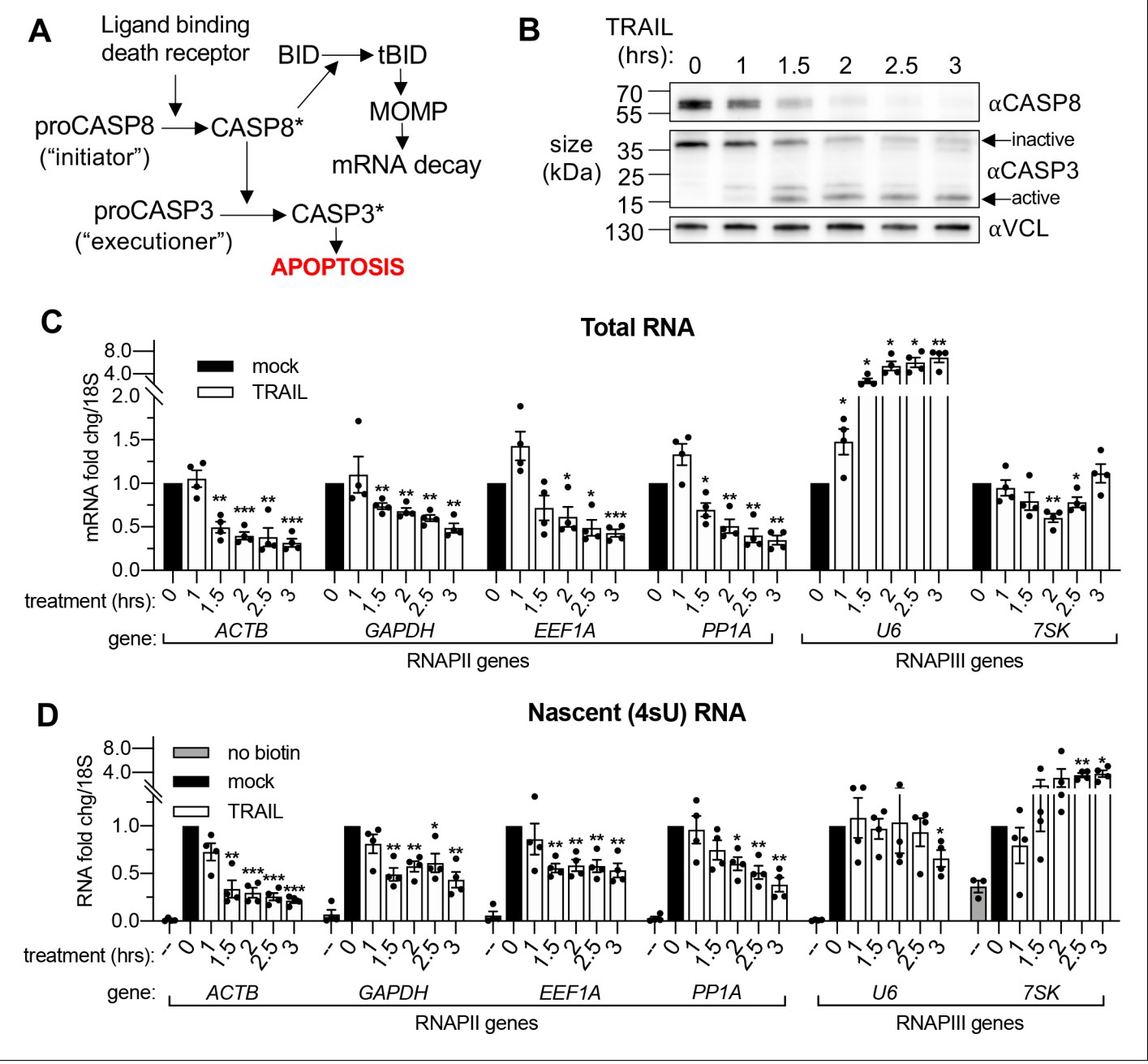

**Figure 1.** mRNA decay during early apoptosis is accompanied by reduced synthesis of RNAPII transcripts. (**A**) Schematic representation of how the extrinsic apoptotic pathway accelerates mRNA decay. (**B**) Western blot of HCT116 lysates showing the depletion of full-length caspase 8 (CASP8) and caspase 3 (CASP3) over a time course of 100 ng/μL TRAIL treatment. Vinculin (VCL) serves as a loading control. Blot representative of those from four biological replicates. (**C, D**) RT-qPCR quantification of total (**C**) and nascent 4sU pulse-labeled (**D**) RNA at the indicated times post TRAIL treatment of HCT116 cells (n = 4). Also see *Figure 1—figure supplement 1*. No biotin control quantifies RNA not conjugated to biotin that is pulled down with strepdavidin selection beads. Fold changes were calculated from $C_t$ values normalized to 18S rRNA in reference to mock treated cells. Graphs display mean ± SEM with individual biological replicates represented as dots. Statistically significant deviation from a null hypothesis of 1 was determined using one sample t test; *p<0.05, **p<0.01, ***p<0.001 (p values provided in *Supplementary file 1A* for all figures).

The online version of this article includes the following figure supplement(s) for figure 1:

**Figure supplement 1.** mRNA decay during early apoptosis is accompanied by reduced synthesis of RNAPII transcripts.

be stable during early apoptosis (*Houge et al., 1995*; *Thomas et al., 2015*). As expected, this decrease was specific to RNAPII transcripts, as the RNA polymerase III (RNAPIII)-transcribed non-coding RNAs (ncRNAs) *U6, 7SK, 7SL, and 5S* did not show a similarly progressive decrease. The *U6* transcript was instead upregulated, possibly suggesting its post-transcriptional regulation as alluded to in a previous study (*Noonberg et al., 1996*). These data confirm that mRNA depletion occurs by 1.5–2 hr during TRAIL-induced apoptosis.

To monitor whether apoptosis also influenced transcription, we pulse labeled the cells with 4-thio-uridine (4sU) for 20 min at the end of each TRAIL treatment. 4sU is incorporated into actively tran-scribing RNA and can be subsequently coupled to HPDP-biotin and purified over streptavidin beads, then quantified by RT-qPCR to measure nascent transcript levels (*Dölken, 2013*). 4sU-labeled RNA levels were also normalized to 18S rRNA, which was produced at a constant level in the pres-ence and absence of TRAIL (*Figure 1—figure supplement 1B*). In addition to a reduction in steady state mRNA abundance, TRAIL treatment caused a decrease in RNAPII-driven mRNA production, while RNAPIII transcription was largely either unaffected or enhanced (*Figure 1D*, *Figure 1—figure supplement 1C*). Thus, TRAIL triggers mRNA decay and decreases nascent mRNA production in HCT116 cells but does not negatively impact RNAPIII transcript abundance or production.

## RNAPII transcription is globally repressed during early apoptosis

z-VAD-fmk (zVAD), a pan-caspase inhibitor, was used to confirm that TRAIL-induced mRNA decay and transcriptional arrest were associated with apoptosis and not due to an off-target effect of TRAIL. HCT116 cells were pre-treated with 40 µM zVAD or equal volume of vehicle (DMSO) for 1 hr prior to TRAIL treatment. The effectiveness of zVAD treatment was confirmed by showing it blocked the cleavage of the CASP8 target BID and blocked the degradation of the CASP3 substrate PARP1 (*Figure 2—figure supplement 1A*). The decreases in total and nascent 4sU-labeled mRNA abun-dance upon TRAIL treatment were rescued in the presence of zVAD (*Figure 2A–B*), confirming the role of canonical apoptotic signaling in both phenotypes.

*Liu et al., 2018* reported that the mRNA degradation occuring shortly after TRAIL treatment in HCT116 cells is widespread. In order to quantify the extent of the associated transcriptional arrest, we sequenced the 4sU-labeled nascent transcriptome of non-apoptotic and apoptotic cells. HCT116 cells were pre-treated with DMSO (i.e. allowing for apoptosis signaling to proceed) or zVAD (to pre-vent apoptosis) prior to addition of TRAIL (or vehicle) for 2 hr. Nascent RNA was depleted of rRNA before sequencing libraries were prepared, and >95% of the resultant reads mapped to mRNAs and other predominantly RNAPII transcripts such as long non-coding RNAs (lncRNAs; *Marchese et al., 2017*) and small nucleolar RNAs (snoRNAs; *Dieci et al., 2009*; *Figure 2—source data 1A–B*). Signifi-cantly more nascent transcripts decreased than increased upon TRAIL treatment in cells pre-treated with DMSO ($p \cong 0$, chi-squared test), while significantly more increased than decreased with TRAIL treatment in the presence of zVAD (p=5.234e-143, chi-squared test). Markedly, 71.2% of the ~28,000 unique transcripts detected were downregulated more than twofold in TRAIL-treated cells with DMSO (*Figure 2C*), while only 14.7% of ~32,000 transcripts were repressed upon TRAIL treatment when caspases were inhibited with zVAD (*Figure 2D*). By contrast, fewer than 20% of tran-scripts were transcriptionally upregulated by the same amount during apoptosis in both conditions (*Figure 2C–D*). To validate that RNA production was accurately captured in the sequencing libraries, fold changes for a representative transcript (*ACTB*) in the original 4sU-labeled RNA samples were assessed by RT-qPCR using both exonic and intronic primers (*Figure 2—figure supplement 1B*), showing good agreement between the three quantification methods.

Of the transcripts changed by twofold or greater in TRAIL-treated cells in the absence of zVAD, 745 decreased and 69 increased in a statistically significant manner across two biological replicates (*Figure 2—figure supplement 1C*). By contrast, only 56 transcripts were significantly downregulated upon TRAIL treatment in the presence of zVAD, while 364 were upregulated (*Figure 2—figure sup-plement 1D*). Gene ontology enrichment analysis revealed that the genes upregulated upon TRAIL treatment alone were disproportionately involved in cellular responses to chemokines and cell death (*Figure 2E*, *Figure 2—source data 1C*), while no statistically significant enrichments were observed in the downregulated transcripts (*Supplementary file 2*). Interestingly, the induced genes were more likely to be regulated by transcription factors implicated in apoptosis such as CSRNP1 (*Ye et al., 2017*), FOSB (*Baumann et al., 2003*), NR4A3 (*Fedorova et al., 2019*), NFKB2 (*Keller et al., 2010*), JUNB (*Gurzov et al., 2008*), JUN (*Wisdom, 1999*), EGR1 (*Pignatelli et al.,*

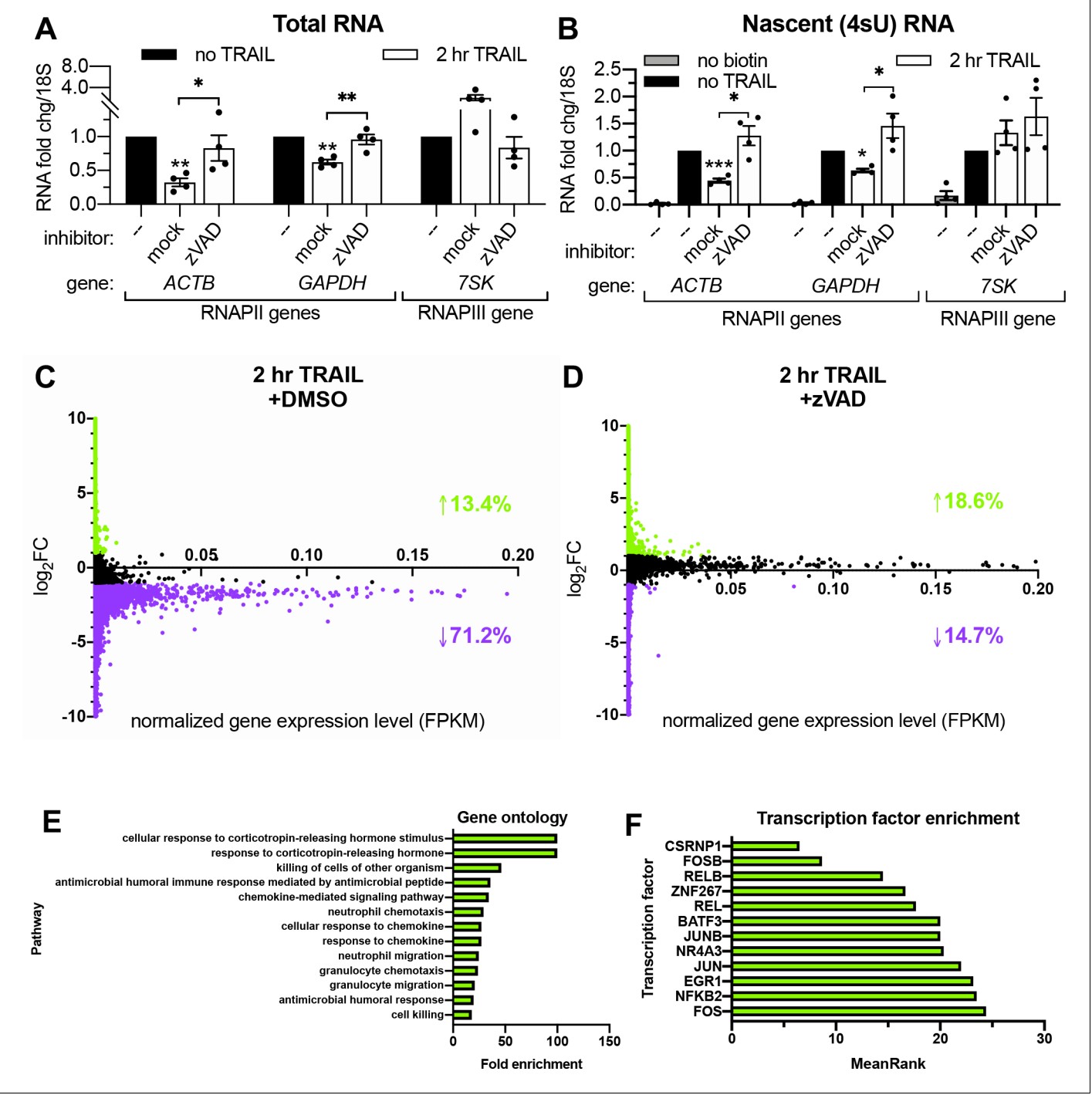

**Figure 2.** RNAPII transcription is globally repressed during early apoptosis. (A, B) RT-qPCR measurements of total (A) and nascent 4sU-labeled (B) RNA fold changes after 2 hr TRAIL treatment of HCT116 cells, including a 1 hr pre-treatment with either 40 µM zVAD or an equal volume of DMSO ('mock'). Also see *Figure 2—figure supplement 1A*. RNA fold change values were calculated in reference to *18S* rRNA. Bar graphs display mean ± SEM with individual biological replicates (*n* = 4) represented as dots. Statistically significant deviation from a null hypothesis of 1 was determined using one sample t test and indicated with asterisks directly above bars, while student's t tests were performed to compare mean fold change values for mock inhibitor or scramble treated cells to those treated with inhibitor or a targeting siRNA and indicated with brackets. *p<0.05, **p<0.00.1, ***p<0.001. (C, D) rRNA-depleted cDNA sequencing libraries were reverse transcribed from 4sU-labeled RNA isolated from cells under the conditions described in (A, B). Transcripts that aligned to genes in the human genome are graphed with differential log$_2$ fold change expression values (log$_2$FC) on the y axis and fragments per kilobase per million reads (FKPM) expression values (normalized to ERCC spike-in controls) on the x axis. All values were averaged from two biological replicates. Data points for transcripts upregulated or downregulated by twofold or greater are colored green and purple, respectively. *Figure 2 continued on next page*

*Figure 2 continued*

Percentages of transcripts in each expression class are indicated with an arrow and in their corresponding colors. Also see *Figure 2—figure supplement 1C–D* and *Figure 2—source data 1A–B*. (E) Top statistically significant hits from gene ontology analysis performed for the list of transcripts that were upregulated upon TRAIL treatment with DMSO in a statistically significant manner across biological duplicates. Also see *Figure 2—source data 1C*. (F) Top hits from transcription factor (TF) enrichment analysis for the same list of genes as above. The lower the MeanRank value, the more statistically significant enrichment for genes regulated by the indicated TF. Also see *Figure 2—source data 1D*.

The online version of this article includes the following source data and figure supplement(s) for figure 2:

**Source data 1.** 4sU-seq differential gene expression and enrichment analyses.

**Figure supplement 1.** RNAPII transcription is globally repressed during early apoptosis.

---

*2003*), FOS (*Preston et al., 1996*), KFL6 (*Huang et al., 2008*), RELB (*Guerin et al., 2002*), and BATF3 (*Qiu et al., 2020*), indicating that the dataset reflects established apoptotic transcriptional dynamics (*Figure 2F*, *Figure 2—source data 1D*).

## Transcriptional repression during early apoptosis requires MOMP, but not necessarily caspase activity

We next sought to determine whether any hallmark features of apoptosis underlay the observed transcriptional repression. These include the limited proteolysis by the 'initiator' CASP8, mass proteolysis by the 'executioner' CASP3, the endonucleolytic cleavage of the genome by a caspase-activated DNase (CAD) that translocates into the nucleus during late stages of apoptosis (*Enari et al., 1998*) and MOMP (which instigates mRNA decay).

Small interfering RNA (siRNA) knockdowns (*Figure 3A*) were performed to first determine if the rescue of mRNA production caused by zVAD was due specifically to the inhibition of the initiator CASP8 or the executioner CASP3. Both accelerated mRNA decay (*Figure 3—figure supplement 1A*) and RNAPII transcriptional repression (*Figure 3B*) upon 2 hr TRAIL treatment required the MOMP-inducing CASP8 but not CASP3, suggesting that mass proteolysis by CASP3 does not significantly contribute to either phenotype. Knockdown of CAD did not affect the reduction in 4sU incorporation observed during early apoptosis (*Figure 3B*), and DNA fragmentation (as measured by TUNEL assay) was not detected in TRAIL-treated HCT116 cells until 4–8 hr post-treatment (*Figure 3—figure supplement 1C*). These findings are in agreement with the prior study showing that MOMP and mRNA decay occur before DNA fragmentation begins during extrinsic apoptosis (*Thomas et al., 2015*).

Although CASP3 is dispensable for apoptotic transcriptional repression, a previous report suggests that CASP8 may cleave RPB1, the largest subunit of RNAPII (*Lu et al., 2002*). We therefore measured the protein expression of RPB1 and the three next largest RNAPII subunits (RBP2-4) during early apoptosis to determine if degradation of these subunits might explain the observed TRAIL-induced reduction in RNAPII transcription. Expression of RPB1-4 was relatively unaffected by 2 hr TRAIL treatment in the presence or absence of zVAD, with the exception of a small decrease in the amount of RPB2 that was rescued upon zVAD treatment (*Figure 3—figure supplement 1C*). Thus, RNAPII depletion is unlikely to underlie the transcriptional repression phenotype.

Our above observations suggest that MOMP activation, which can occur through CASP8, is necessary to drive apoptotic mRNA decay and the ensuing transcriptional repression. To test this hypothesis, we attenuated MOMP in TRAIL-treated cells by depleting the mitochondrial pore-forming proteins BAX and BAK (*Figure 3C*). Indeed, siRNA-mediated depletion of BAX and BAK rescued cytoplasmic mRNA abundance (*Figure 3—figure supplement 1D*) and RNAPII transcription (*Figure 3D*) of the *ACTB* and *GAPDH* transcripts in the presence of TRAIL, even though CASP8 and CASP3 were still cleaved (*Figure 3D*). Thus, CASP8 likely participates in this pathway only to the extent that it activates MOMP, as MOMP appears to be the main driver of mRNA decay and transcriptional repression.

Finally, to confirm that MOMP is sufficient to drive this phenotype, we used a small molecule, raptinal, that bypasses CASP8 to intrinsically induce MOMP (*Heimer et al., 2019*; *Palchaudhuri et al., 2015*). HeLa cells treated with 10 μM raptinal for 4 hr underwent MOMP, as measured by cytochrome c release into the cytoplasm, in the presence or absence of zVAD (*Figure 3—figure supplement 1E*). Steady state levels (*Figure 3E*) and transcription (*Figure 3F*) of the aforementioned

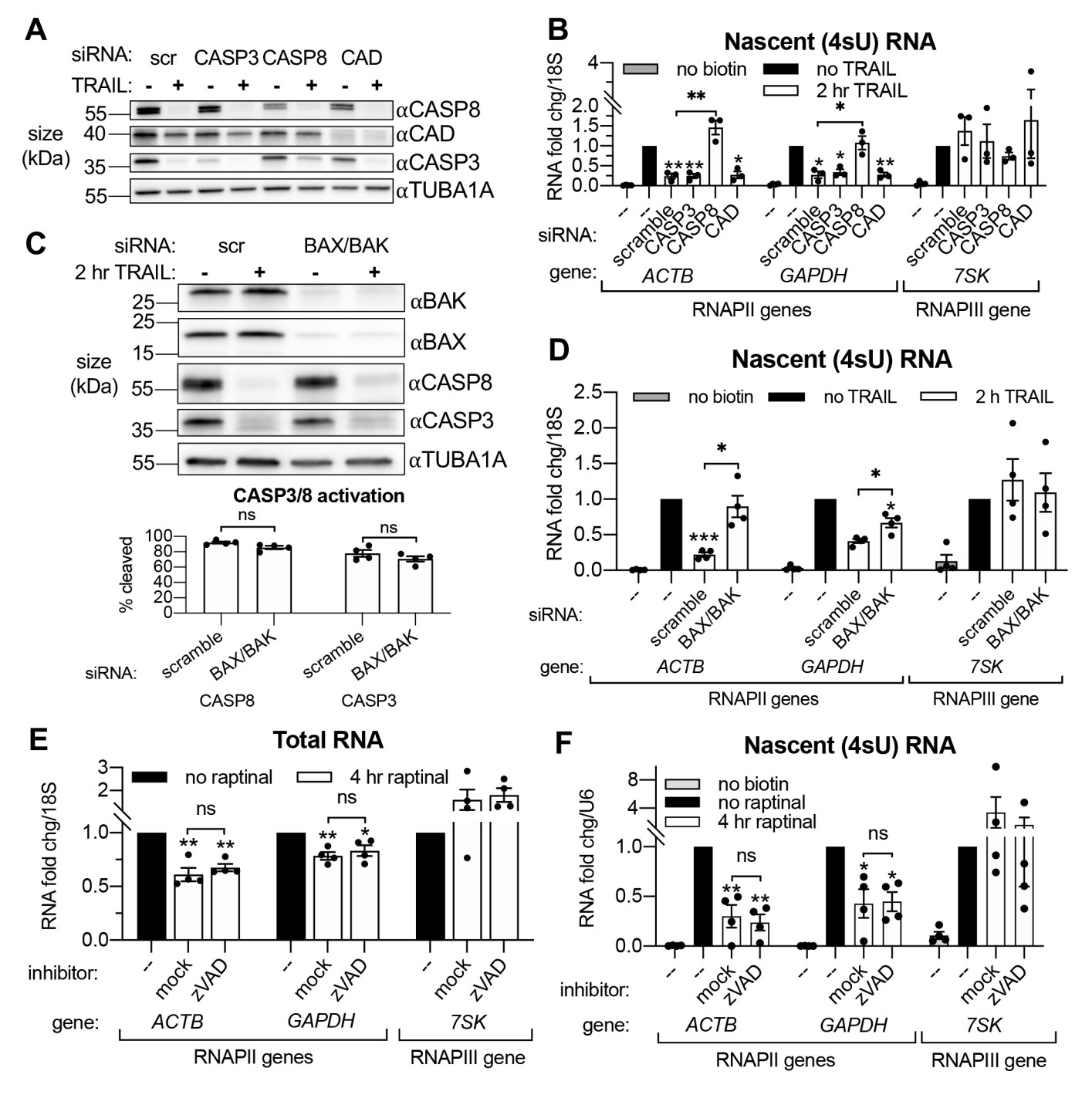

**Figure 3.** Transcriptional repression during early apoptosis requires MOMP, but not necessarily caspase activity. (**A**) Western blots showing the efficacy of CASP3, CASP8, and caspase-activated DNase (CAD) protein depletion following nucleofection with the indicated siRNAs, with or without 2 hr TRAIL treatment. α-Tubulin (TUBA1A) serves as a loading control. Blot representative of those from four biological replicates. (**B**) RT-qPCR measurements of 4sU-labeled nascent transcripts with or without 2 hr TRAIL treatment in cells nucleofected with the indicated the siRNAs (n = 3). Also see ***Figure 3—figure supplement 1A–C***. (**C**) Western blot detecting the indicated proteins in cells nucleofected with the indicated siRNA in the presence or absence of TRAIL. TUBA1A serves as a loading control. Blot representative of four biological replicates. The cleavage of CASP3 and CASP8 (as measured by the disappearance of the full-length form of each zymogen upon 2 hr TRAIL treatment, normalized to TUBA1A) is graphed below (n = 4). (**D**) 4sU-labeled RNA levels measured by RT-qPCR in HCT116 cells nucleofected with the indicated siRNAs, with or without 2 hr TRAIL treatment (n = 4). Also see ***Figure 3—figure supplement 1D***. (**E**, **F**) Total (**E**) and 4sU-labeled (**F**) RNA levels measured by RT-qPCR in HeLa cells after 10 µM raptinal treatment for 4 hr, with or without a 1 hr pre-treatment of 20 µM zVAD (n = 4). Also see ***Figure 2—figure supplement 1E***. Fold changes were calculated in reference

*Figure 3 continued on next page*

*Figure 3 continued*

to the *U6* small nuclear RNA (snRNA) transcript since its production was more stable after 4 hr raptinal than that of *18S* rRNA (see ***Figure 2—figure supplement 1F***). All RNA fold changes were calculated from $C_t$ values normalized to *18S* or *U6* RNA, then normalized to non-apoptotic cells ('no TRAIL') under otherwise identical conditions. Graphs display mean ± SEM with individual biological replicates represented as dots. Statistically significant deviation from a null hypothesis of 1 was determined using one sample t test and indicated with asterisks directly above bars, while student's t tests were performed to compare mean fold change values for mock inhibitor or scramble treated cells to those treated with zVAD or a targeting siRNA and indicated with brackets. The Holm-Sidak correction for multiple comparisons was applied in the student's t tests represented in (A, B) \*p<0.05, \*\*p<0.00.1, \*\*\*p<0.001.

The online version of this article includes the following figure supplement(s) for figure 3:

**Figure supplement 1.** Transcriptional repression during early apoptosis requires MOMP, but not necessarily caspase activity.

mRNAs was reduced upon raptinal treatment. The fact that this reduction in mRNA abundance and synthesis was maintained upon caspase inhibition by zVAD treatment indicates that the caspases are not required to drive these phenotypes outside of their role in MOMP activation. Taken together, these data confirm that the mRNA degradation and ensuing transcriptional repression observed during early apoptosis are driven by MOMP.

## Cytoplasmic 3'- but not 5'-RNA exonucleases are required for apoptotic RNAPII transcriptional repression

Based on connections between virus-activated mRNA decay and RNAPII transcription (***Abernathy et al., 2015***; ***Gilbertson et al., 2018***), we hypothesized that TRAIL-induced mRNA turnover was functionally linked to the concurrent transcriptional repression. Apoptotic mRNA decay occurs from the 3' end by the actions of the cytoplasmic 3'-RNA exonuclease DIS3L2 and the mitochondrial 3'-RNA exonuclease PNPT1, which is released into the cytoplasm by MOMP (***Liu et al., 2018***; ***Thomas et al., 2015***). This stands in contrast to basal mRNA decay, which occurs predominantly from the 5' end by XRN1 (***Jones et al., 2012***). We therefore set out determine if 3' or 5' decay factors were required for apoptosis-linked mRNA decay and the ensuing repression of mRNA transcription. Depletion of DIS3L2, PNPT1, or the cytoplasmic 3' RNA exosome subunit EXOSC4 individually did not reproducibly rescue the total levels of either the *ACTB* or *GAPDH* mRNA during early apoptosis (***Figure 4—figure supplement 1A***), nor did they affect the relative production of these transcripts (***Figure 4—figure supplement 1B***). Given the likely redundant nature of the multiple 3' end decay factors (***Houseley and Tollervey, 2009***), we instead performed concurrent knockdowns of DIS3L2, EXOSC4, and PNPT1 to more completely inhibit cytoplasmic 3' RNA decay. We also knocked down the predominant 5'−3' RNA exonuclease XRN1 to check the involvement of 5' decay (***Figure 4A***). Depletion of the 3'−5' but not the 5'−3' decay machinery attenuated the apoptotic decrease in total RNA levels and largely restored RNAPII transcription (***Figure 4B–C***). Importantly, there was only a minor reduction in CASP3 activation in cells depleted of 3'−5' decay factors and this was not significantly different from that observed upon XRN1 knockdown (***Figure 4A***). These observations suggest that decreased RNAPII transcription occurs as a consequence of accelerated 3' mRNA degradation in the cytoplasm during early apoptosis.

## Apoptosis causes reduced RNAPII promoter occupancy in an mRNA decay-dependent manner

RNAPII is recruited to promoters in an unphosphorylated state, but its subsequent promoter escape and elongation are governed by a series of phosphorylation events in the heptad $(Y_1S_2P_3T_4S_5P_6S_7)_n$ repeats of the RPB1 C-terminal domain (CTD). To determine the stage of transcription impacted by mRNA decay, we performed RNAPII ChIP-qPCR and western blots using antibodies recognizing the different RNAPII phosphorylation states. Occupancy of hypophosphorylated RNAPII at the *ACTB* and *GAPDH* promoters was significantly reduced after 2 hr TRAIL treatment (***Figure 3D***). In accordance with the 4sU labeling results, siRNA-mediated knockdowns showed that loss of RNAPII occupancy in response to TRAIL requires CASP8 but not CASP3 (***Figure 4—figure supplement 1C***), as well as cytoplasmic 3'−5' RNA decay factors but not the 5'−3' exonuclease XRN1 (***Figure 4D***). Impaired binding of the TATA-binding protein (TBP), which nucleates the formation of the RNAPII pre-initiation complex (PIC) at promoters (***Buratowski et al., 1989***; ***Louder et al., 2016***), mirrored

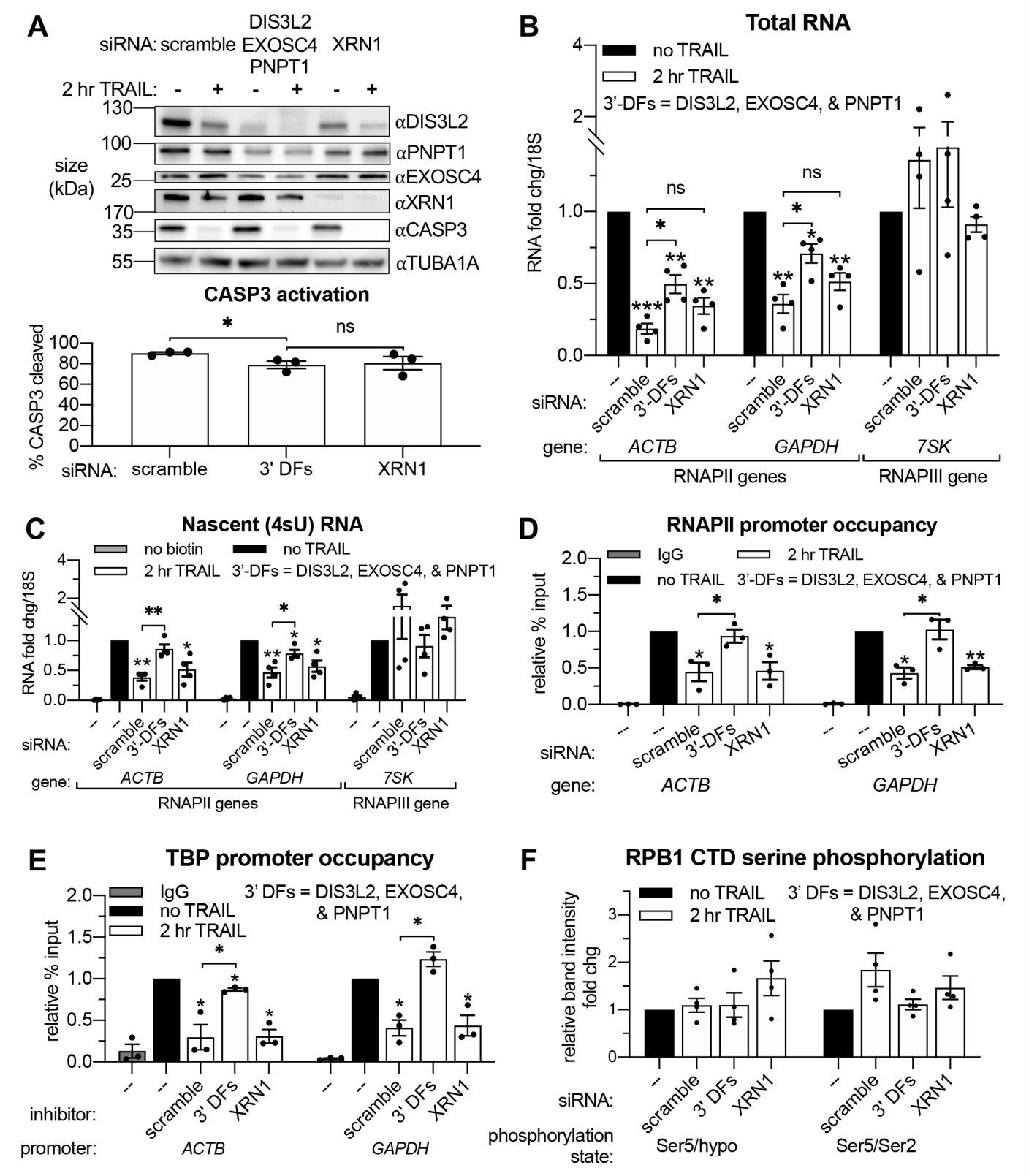

**Figure 4.** Apoptosis causes reduced RNAPII transcriptional output and promoter occupancy in an mRNA decay-dependent manner. (**A**) Western blots performed with lysates from HCT116 cells depleted of the indicated decay factors with and without 2 hr TRAIL treatment. Blot representative of three biological replicates. Apoptosis induction was confirmed by disappearance of the full-length CASP3 band, quantified in the graph below by measuring

*Figure 4 continued on next page*

*Figure 4 continued*

band intensity normalized to an α-tubulin (TUBA1A) loading control (*n* = 3). (**B, C**) Changes in total (**B**) and nascent 4sU-labeled (**C**) RNA upon 2 hr TRAIL treatment in cells nucleofected with the indicated siRNAs were quantified by RT-qPCR (*n* = 4). Fold changes were calculated from $C_t$ values normalized to *18S* rRNA. Also see *Figure 4—figure supplement 1A–B*. (**D, E**) Chromatin immunoprecipitation (ChIP)-qPCR was used to measure occupancy of the indicated promoters by hypophosphorylated RNAPII (**D**) or TBP (**E**) under cellular conditions described in (**A**). Rabbit and mouse IgG antibodies were included in parallel immunoprecipitation reactions with chromatin from scramble siRNA-treated non-apoptotic cells in lieu of TBP and RNAPII antibodies, respectively, as a control. Also see *Figure 4—figure supplement 1D–E*. (**F**) Relative band intensity ratios from four replicates of the representative western blots depicted in *Figure 4—figure supplement 1D*, using primary antibodies specific to the indicated RPB1 CTD phosphorylation state under cellular conditions described in (**A**). Band intensity values were first normalized to a vinculin (VCL) loading control in each blot. All bar graphs display mean ± SEM with individual biological replicates represented as dots. Statistically significant deviation from a null hypothesis of 1 was determined using one sample t test and indicated with asterisks directly above bars, while student's t tests with the Holm-Sidak correction for multiple comparisons were performed to compare mean values between groups indicated with brackets. *p<0.05, **p<0.00.1, ***p<0.001.

The online version of this article includes the following figure supplement(s) for figure 4:

**Figure supplement 1.** Apoptosis causes reduced RNAPII transcriptional output and promoter occupancy in an mRNA decay-dependent manner.

that of RNAPII (*Figure 4E*). These changes are not driven by alterations in the stability of RPB1 or TBP, since the expression of these proteins remained relatively constant during early apoptosis regardless of the presence of mRNA decay factors (*Figure 4—figure supplement 1D–E*). The relative proportion of initiating RPB1 CTD phosphorylated at the serine five position and the ratio of serine 5 to serine 2 phosphorylation, which decreases during transcriptional elongation (*Shandilya and Roberts, 2012*), also remain unchanged during early apoptosis regardless of the presence of mRNA decay factors (*Figure 4F*). These data suggest that the decay-dependent reduction in mRNA synthesis occurs at or before PIC formation, rather than during transcriptional initiation and elongation.

## Importin α/β transport links mRNA decay and transcription

Finally, we sought to determine how TRAIL-induced cytoplasmic mRNA decay signals to the nucleus to induce transcriptional repression. Data from viral systems suggest that this signaling involves differential trafficking of RNA-binding proteins (RBPs), many of which transit to the nucleus in response to virus-induced cytoplasmic mRNA decay (*Gilbertson et al., 2018*). We therefore sought to test the hypothesis that nuclear import of RBPs underlies how cytoplasmic mRNA decay is sensed by the RNAPII transcriptional machinery.

To determine whether RBP redistribution occurs during early apoptosis, we analyzed the subcellular distribution of cytoplasmic poly(A) binding protein PABPC1, an RBP known to shuttle to the nucleus in response to virus-induced RNA decay (*Gilbertson et al., 2018*; *Kumar and Glaunsinger, 2010*; *Lee and Glaunsinger, 2009*). We performed cell fractionations of HCT116 cells to measure PABPC1 levels in the nucleus versus the cytoplasm upon induction of apoptosis. Indeed, 2 hr after TRAIL treatment, PABPC1 protein levels increased in the nuclear fraction of cells but not in the cytoplasmic fraction, indicative of relocalization (*Figure 5A*). This increase in nuclear PABPC was dependent on the presence of CASP8 but not CASP3, mirroring the incidence of mRNA decay and reduced RNAPII transcription. Thus, similar to viral infection, PABPC1 relocalization occurs coincidentally with increased mRNA decay and transcriptional repression in the context of early apoptosis.

We next sought to directly evaluate the role of protein shuttling in connecting cytoplasmic mRNA decay to transcriptional repression. The majority of proteins ~ 60 kilodaltons (kDa) and larger cannot passively diffuse through nuclear pores; they require assistance by importins to enter the nucleus from the cytoplasm (*Görlich, 1998*). Classical nuclear transport occurs by importin α binding to a cytoplasmic substrate, which is then bound by an importin β to form a tertiary complex that is able to move through nuclear pore complexes (*Stewart, 2007*). To test if this mode of nuclear transport is required for transcriptional feedback, HCT116 cells were pretreated with ivermectin, a specific inhibitor of importin α/β transport (*Wagstaff et al., 2012*), before the 2 hr TRAIL treatment. The efficacy of ivermectin was validated by an observed decrease in the nuclear levels of the RBP nucleolin (NCL), a known importin α/β substrate (*Kimura et al., 2013*; *Figure 5*). Ivermectin pretreatment rescued RNAPII transcription upon TRAIL treatment (*Figure 5C*) without significantly affecting the extent of mRNA decay (*Figure 5—figure supplement 1A*), suggesting that protein trafficking to the nucleus provides signals connecting cytoplasmic mRNA decay to transcription.

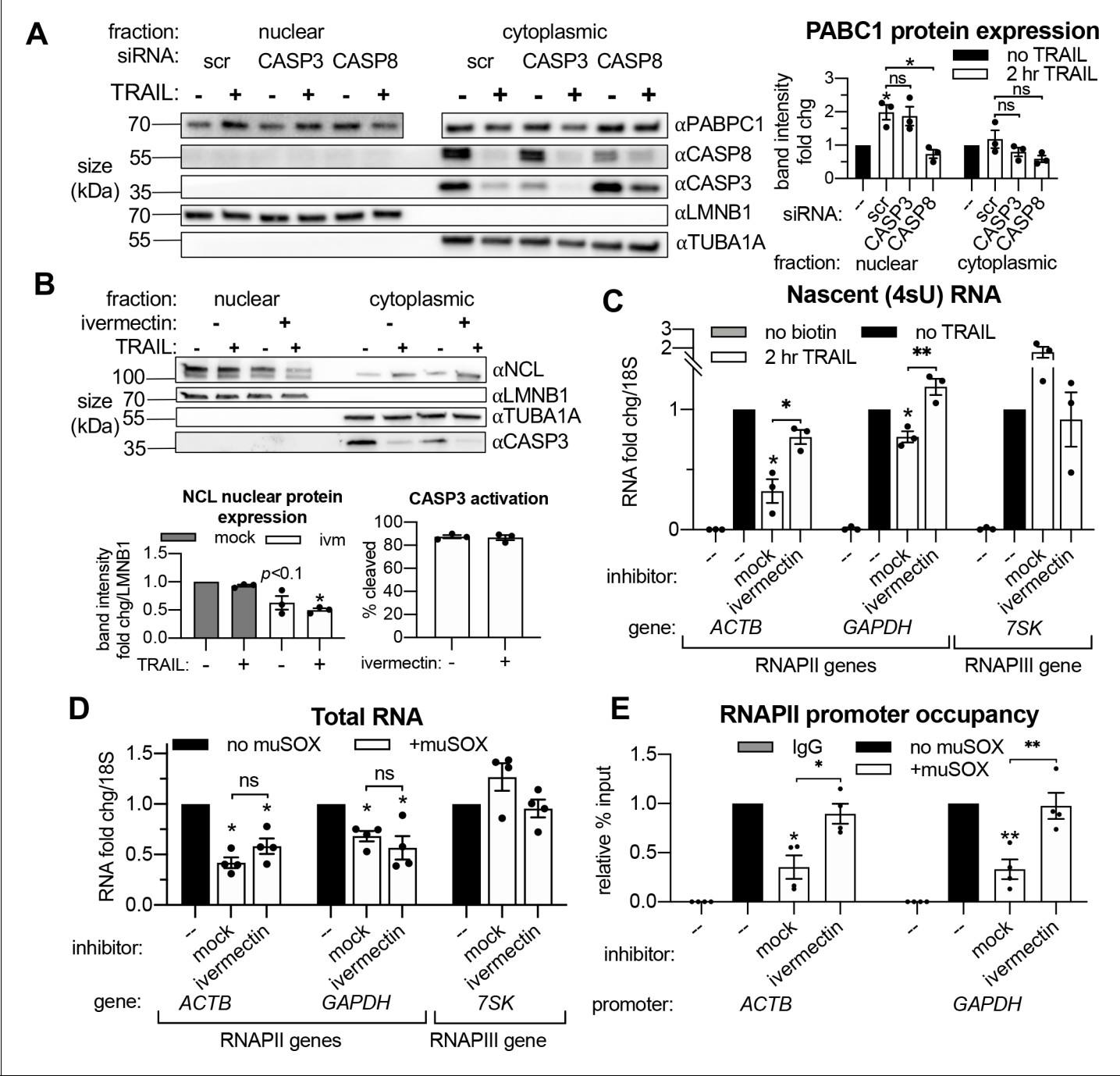

**Figure 5.** Importin α/β transport links mRNA decay and transcription. (**A**) Western blots showing the indicated proteins in the nuclear and cytoplasmic fractions of apoptotic and non-apoptotic HCT116 cells nucleofected with the indicated siRNAs. The nuclear and cytoplasmic fractions of PABPC1 were imaged on the same membrane section but cropped and edited separately to visualize the low nuclear expression of this canonically cytoplasmic protein. Protein expression (right) was calculated by band intensity in reference to lamin B1 (LMNB1) or α-tubulin (TUBA1A) loading controls, for the nuclear and cytoplasmic fractions, respectively (n = 3). (**B**) Nuclear and cytoplasmic expression of the indicated proteins in apoptotic and non-apoptotic cells with a 1 hr pre-treatment with 25 μM ivermectin or an equal volume of EtOH ('mock'). Nuclear levels of the importin α/β substrate nucleolin (NCL) were quantified by band intensity in reference to a LMNB1 loading control, while disappearance of full-length CASP3 was quantified in the cytoplasm in reference to TUBA1A (n = 3). Also see *Figure 5—figure supplement 1A* (**C**) Levels of nascent 4sU-labeled RNA (n = 3) were quantified by RT-qPCR under the cellular conditions described in (**B**). Also see *Figure 5—figure supplement 1B–C*. (**D**) RT-qPCR quantification of total RNA levels in HEK293T cells stably expressing a doxycycline (dox)-inducible form of muSOX endonuclease, cultured with or without 1 μg/mL dox for 24 hr. Cells were treated with 25 μM ivermectin or an equal volume of EtOH 2 hr before harvesting (n = 4). (**E**) RNAPII promoter occupancy at the ACTB and GAPDH promoters (n = 4) was determined by ChIP-qPCR under cellular conditions described in (**D**). Also see *Figure 5—figure supplement 1D–F*. RNA fold

*Figure 5 continued on next page*

*Figure 5 continued*

changes were calculated from $C_t$ values normalized to 18S rRNA. All bar graphs display mean ± SEM with individual biological replicates represented as dots. Statistically significant deviation from a null hypothesis of 1 was determined using one sample t test and indicated with asterisks directly above bars, while student's t tests were performed to compare mean fold change values for mock inhibitor or scramble treated cells to those treated with ivermectin or a targeting siRNA and indicated with brackets. The Holm-Sidak correction for multiple comparisons was applied in the student's t tests represented in (**A**). *p<0.05, **p<0.00.1, ***p<0.001.

The online version of this article includes the following figure supplement(s) for figure 5:

**Figure supplement 1.** Importin α/β transport links mRNA decay and transcription.

Importantly, ivermectin did not diminish the extent of CASP3 cleavage during early apoptosis (*Figure 5B*), nor did it decrease baseline levels of transcription in non-apoptotic cells (*Figure 5—figure supplement 1B*). Interestingly, it also did not prevent PABPC1 import (*Figure 5—figure supplement 1C*), suggesting that PABPC1 translocation likely occurs via an importin α/β-independent pathway and is not sufficient to repress RNAPII transcription.

Finally, we evaluated whether importin α/β transport is also required for feedback between viral nuclease-driven mRNA decay and RNAPII transcription, as this would suggest that the underlying mechanisms involved in activating this pathway may be conserved. We used the mRNA-specific endonuclease muSOX from the gammaherpesvirus MHV68, as muSOX expression has been shown to cause widespread mRNA decay and subsequent transcriptional repression (*Abernathy et al., 2014*; *Abernathy et al., 2015*). HEK-293T cell lines were engineered to stably express dox-inducible wild-type muSOX or the catalytically inactive D219A mutant (*Abernathy et al., 2015*). These cells were treated with ivermectin for 3 hr and the resultant changes in RNAPII promoter occupancy were measured by ChIP-qPCR. As expected, expression of WT (*Figure 5D–E*) but not D219A muSOX (*Figure 5—figure supplement 1E–F*) caused mRNA decay and transcriptional repression. Notably, inhibiting nuclear import with ivermectin (*Figure 5—figure supplement 1D*) rescued RNAPII promoter occupancy (*Figure 5E*) without altering the extent of mRNA decay in muSOX expressing cells (*Figure 5D*). Thus, importin α/β nuclear transport plays a key role in linking cytoplasmic mRNA decay to nuclear transcription, both during early apoptosis and upon viral nuclease expression.

## Discussion

mRNA decay and synthesis rates are tightly regulated in order to maintain appropriate levels of cellular mRNA transcripts (*Braun and Young, 2014*). It is well established that when cytoplasmic mRNA is stabilized, for example by the depletion of RNA exonucleases, transcription often slows in order to compensate for increased transcript abundance (*Haimovich et al., 2013*; *Helenius et al., 2011*; *Singh et al., 2019*; *Sun et al., 2012*). Eukaryotic cells thus have the capacity to 'buffer' against reductions in mRNA turnover or synthesis. Here, we revealed that a buffering response does not occur under conditions of elevated cytoplasmic mRNA degradation stimulated during early apoptosis. Instead, cells respond to cytoplasmic mRNA depletion by decreasing RNAPII promoter occupancy and transcript synthesis, thereby amplifying the magnitude of the gene expression shut down. Nuclear import of cytoplasmic proteins is required for this 'transcriptional feedback', suggesting a pathway of gene regulation in which enhanced mRNA decay prompts cytoplasmic proteins to enter the nucleus and halt mRNA production. Notably, similar transcriptional feedback is elicited during virus-induced mRNA decay (*Abernathy et al., 2015*; *Gilbertson et al., 2018*; *Hartenian et al., 2020*), indicating that distinct cellular stresses can converge on this pathway to potentiate a multi-tiered shutdown of gene expression.

Multiple experiments support the conclusion that the TRAIL-induced transcriptional repression phenotype is a consequence cytoplasmic decay triggered by MOMP, rather than caspase activity. XRN1-driven 5′−3′ end decay is the major pathway involved in basal mRNA decay (*Łabno et al., 2016*), but MOMP-induced mRNA decay is primarily driven by 3′ exonucleases such as PNPT1 and DIS3L2 (*Liu et al., 2018*; *Thomas et al., 2015*). Accordingly, co-depletion of 3′ decay factors but not XRN1 restored RNAPII promoter occupancy and transcription during early apoptosis. In contrast to that of 3′ mRNA decay factors, depletion of CASP3 (or the caspase activated DNase CAD) did not block mRNA degradation or transcriptional repression, even though CASP3 is responsible for the vast majority of proteolysis that is characteristic of apoptotic cell death (*Walsh et al., 2008*).

Additionally, the initiator CASP8 was required only under conditions where its activity was needed to induce MOMP. These findings reinforce the idea that mRNA decay and transcriptional repression are very early events that are independent of the subsequent cascade of caspase-driven phenotypes underlying many of the hallmark features of apoptosis.

A key open question is what signal conveys cytoplasmic mRNA degradation information to the nucleus to cause transcriptional repression. Our data are consistent with a model in which the signal is provided by one or more proteins imported into the nucleus in response to accelerated mRNA decay (*Figure 6*). Indeed, many cytoplasmic RNA binding proteins undergo nuclear-cytoplasmic redistribution under conditions of viral nuclease-induced mRNA decay, including PABPC (*Gilbertson et al., 2018*; *Kumar and Glaunsinger, 2010*; *Kumar et al., 2011*). We propose that a certain threshold of mRNA degradation is necessary to elicit protein trafficking and transcriptional repression. Presumably, normal levels of basal mRNA decay and regular cytoplasmic repopulation result in a balanced level of mRNA-bound versus unbound proteins. However, if this balance is tipped during accelerated mRNA decay, an excess of unbound RNA binding proteins could accumulate and be transported into the nucleus.

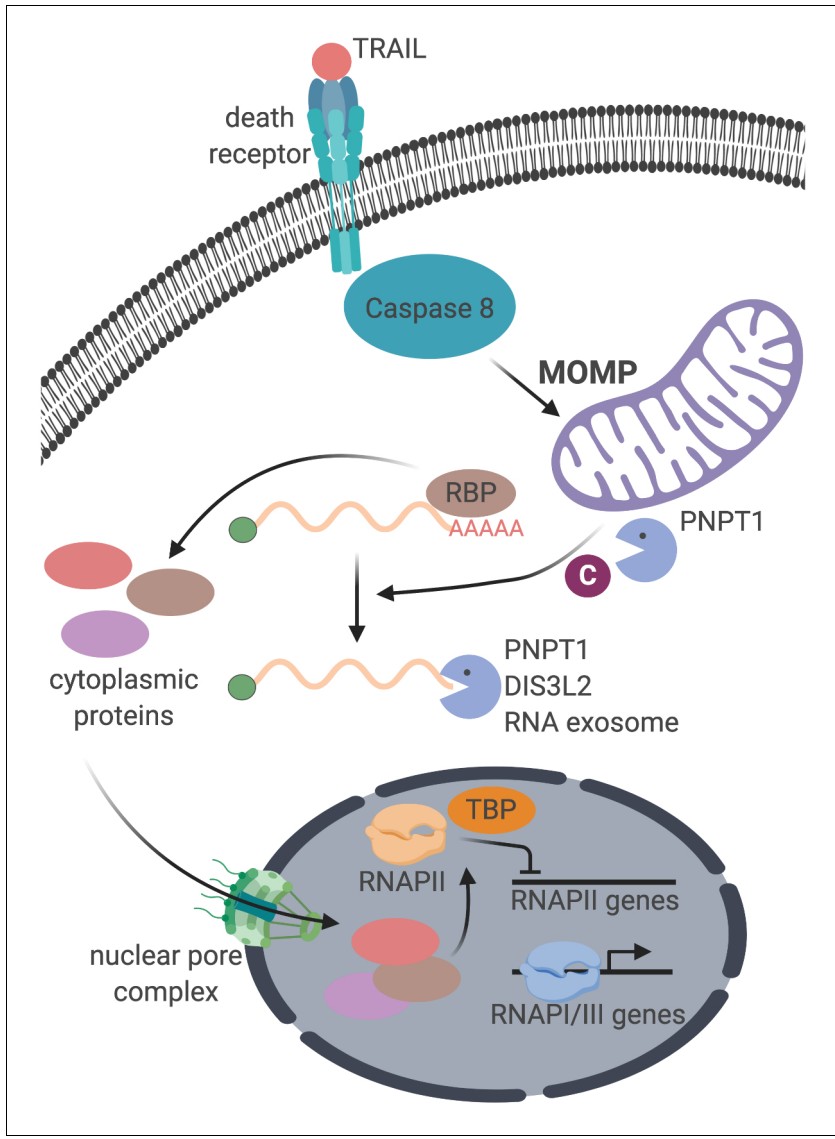

**Figure 6.** Schematic representation of the cellular events connecting apoptosis induction with mRNA decay and RNAPII transcription.

PABPC1 is likely among the first proteins released from mRNA transcripts undergoing 3' end degradation, and its nuclear translocation is driven by a poly(A)-masked nuclear localization signal that is exposed upon RNA decay (*Kumar et al., 2011*). Indeed, we found that PABPC1 undergoes nuclear translocation during early apoptosis. However, the fact that ivermectin blocks transcriptional repression but not PABPC1 import indicates that while PABPC1 redistribution is a marker of mRNA decay, it is not sufficient to induce transcriptional repression in this system. Instead, we hypothesize that the observed RNAPII transcriptional repression occurs as a result of the cumulative nuclear import of multiple factors via importin α/β. Our group previously reported that at least 66 proteins in addition to PABPC1 are selectively enriched in the nucleus upon transfection with the viral endonuclease muSOX (but not with the catalytically-dead D219A mutant), 22 of which are known to be RNA-binding proteins and 45 of which shuttle in a manner dependent on the cytoplasmic mRNA exonuclease primarily responsible for clearing endonuclease cleavage fragments, XRN1 (*Gilbertson et al., 2018*). Future studies in which changes in the nuclear and cytoplasmic proteome upon apoptosis induction and viral endonuclease expression in the presence and absence of ivermectin will likely provide insight into which additional protein or proteins play a role in connecting cytoplasmic mRNA turnover to RNAPII transcription. Nonetheless, the requirement for importin α/β-mediated nuclear transport in both apoptosis-induced and viral nuclease-induced transcriptional repression suggests that these two stimuli may activate a conserved pathway of gene regulation. Whether other types of cell stress elicit a similar response remains an important question for future investigation.

The mRNA 3' end decay-dependent decrease in RNA synthesis is accompanied by a reduction in TBP and RNAPII occupancy at promoters, consistent with transcriptional inhibition occurring upstream of RNAPII initiation and elongation (*Gilbertson et al., 2018*; *Hartenian et al., 2020*). The mechanism driving such a defect is yet to be defined, but possibilities include changes to transcript processing pathways, transcriptional regulators, or the chromatin state that directly or indirectly impact formation of the preinitiation complex. In this regard, mRNA processing and transcription are functionally linked (*Bentley, 2014*) and nuclear accumulation of PABPC1 has been shown to affect mRNA processing by inducing hyperadenylation of nascent transcripts (*Kumar and Glaunsinger, 2010*; *Lee and Glaunsinger, 2009*). Furthermore, TBP interacts with the cleavage–polyadenylation specificity factor (*Dantonel et al., 1997*), providing a possible link between RNAPII preinitiation complex assembly and polyadenylation. Such a mechanism could also explain the specificity of the observed transcriptional repression to RNAPII transcripts. Chromatin architecture could also be influenced by shuttling of RNA binding proteins, as for example the yeast nucleocytoplasmic protein Mrn1 is implicated in the function of chromatin remodeling complexes (*Düring et al., 2012*). Interestingly, the polycomb repressive chromatin complex 2 (PRC2) and DNA methyltransferase 1 (DNMT1) have been shown to bind both nuclear mRNA and chromatin in a mutually exclusive manner (*Beltran et al., 2016*; *Di Ruscio et al., 2013*; *Garland et al., 2019*), evoking the possibility that a nuclear influx of RBPs could secondarily increase the chromatin association of such transcriptional repressors. Future work will test these possibilities in order to mechanistically define the connection between nuclear import and RNAPII transcription under conditions of enhanced mRNA decay.

There are several potential benefits to dampening mRNA transcription in response to accelerated mRNA turnover. Debris from apoptotic cells is usually cleared by macrophages, but inefficient clearance of dead cells can lead to the development of autoantibodies to intracellular components such as histones, DNA, and ribonucleoprotein complexes. This contributes to autoimmune conditions such as systemic lupus erythematosus (*Caruso and Poon, 2018*; *Nagata et al., 2010*). In the context of infection, many viruses require the host RNAPII transcriptional machinery to express viral genes (*Harwig et al., 2017*; *Rivas et al., 2016*; *Walker and Fodor, 2019*). It may therefore be advantageous for the cell to halt transcription in attempt to pre-empt a viral takeover of mRNA synthesis. That said, as with antiviral translational shutdown mechanisms (*Walsh et al., 2013*), viruses have evolved strategies to evade transcriptional repression (*Hartenian et al., 2020*; *Harwig et al., 2017*), as well as inhibit cell death via apoptosis and/or co-opt apoptotic signaling cascades (*Suffert et al., 2011*; *Tabtieng et al., 2018*; *Zhang et al., 2013*; *Zhou et al., 2017*). In either case, the shutdown of transcription in a cell under stress may protect surrounding cells from danger, particularly in an in vivo context. If true, this type of response is likely to be more relevant in multicellular compared with unicellular organisms.

## Materials and methods

### Lead contact
Further information and requests for resources and reagents should be directed to and will be fulfilled by the Lead Contact, Britt Glaunsinger (glaunsinger@berkeley.edu).

### Materials availability
The dox-inducible muSOX-expressing cell line and rabbit polyclonal anti-muSOX antibody generated in this study are both available upon request.

### Code availability
Sequencing data generated in this study are publicly available on GEO repository (accession number GSE163923).

### Cells and culture conditions
Wild-type HCT116, HEK293T, and HeLa cells (all from ATCC) were obtained from the UC Berkeley Tissue Culture Facility. Cell lines were authenticated by STR analysis and determined to be free of mycoplasma by PCR screening. HEK293Ts were made to stably express doxycycline(dox)-inducible wild-type muSOX and its catalytically-dead D219A mutant by PCR amplifying the aforementioned coding sequences from Addgene plasmids 131702 and 131704 using the muSOX F/R primers (see Key Resources Table) and InFusion cloning these fragments into the Lenti-X Tet-One Inducible Expression System digested with AgeI. Lentivirus was made for both constructs by transfecting HEK293T cells with second generation packaging plasmids and spinfected onto HEK293T cells at a low multiplicity of infection (MOI) as previously described (*Hartenian et al., 2020*). Twenty-four hr later, 350 µg/ml zeocin was added to select for transduced cells. HEK293T and HeLa cells were grown in Dulbecco's Modified Eagle Medium (ThermoFisher Scientific) supplemented with 10% fetal bovine serum (FBS) and 1 U/mL penicillin-streptomycin (pen-strep). HCT116 cells were maintained in McCoy's (modified) 5A medium (ThermoFisher Scientific) with 10% FBS and 1 U/mL pen-strep. All cells were incubated at 37°C with 5% $CO_2$. Cells were maintained in culture in 10 $cm^2$ plates and 1 × $10^6$ cells were seeded into six well plates for all experiments except for chromatin immunoprecipitations, in which 5 × $10^6$ cells were seeded into 10 $cm^2$ plates, and 4sU-sequencing, in which 5 × $10^6$ cells were seeded into 15 $cm^2$ plates.

### siRNA nucleofections
Protein knockdowns were performed using the Neon Transfection System with siRNA pools for the following targets: non-targeting control pool (scramble or scr siRNA), CASP3, CASP8, DFFB, DIS3L2, EXOSC4, PNPT1, XRN1, BAX, and BAK. For all siRNA pools, cells were nucleofected according to manufacturer protocols for HCT116 cells and immediately seeded into plates containing media supplemented with 10% FBS but lacking pen-strep to improve cell viability post-nucleofection. A 50 nM final siRNA concentration was used for pools targeting CASP3, CASP8, and DFFB (CAD), while 200 nM was used for individual knockdown of XRN1, DIS3L2, EXOSC4, and PNPT1, as well as the concurrent knockdowns of DIS3L2/EXOSC4/PNPT1 (66.7 nM each) and BAX/BAK (100 nM each). For all experiments involving siRNA knockdowns, cells were treated and/or harvested once 80–90% confluent in each plate or well, approximately 72 hr post-nucleofection, and protein knockdown was confirmed by western blot. Cell populations of siRNA-transfected cells were split into two wells or plates 24 hr pre-harvesting to allow for the direct comparison between apoptotic and non-apoptotic cells in the same genetic background.

### Apoptosis induction
A total of 100 ng/mL TNF-related apoptosis-inducing ligand (TRAIL) was used to induce rapid extrinsic apoptosis in HCT116 cells. Where indicated, HCT116 cells were pre-treated for 1 hr with 40 µM caspase Inhibitor Z-VAD-FMK (zVAD) or 25 µM ivermectin before TRAIL treatment. Extrinsic apoptosis induction was confirmed by observing CASP8 and CASP3 cleavage on western blots. CASP3/8 cleavage was quantified by disappearance in intensity of the full-sized band normalized to TUBA1A or VCL using Bio-Rad ImageLab software. Intrinsic apoptosis was induced in HeLa cells with a 4 hr

treatment of 10 µM raptinal, with or without 1 hr pre-treatment with 20 µM zVAD. Induction was confirmed by evidence of cytochrome c release into the cytoplasm with cell fractionation and western blot. 'Mock' treatments consisted of an equal volume of vehicle used to dissolve each reagent: TRAIL storage and dilution buffer, DMSO, and ethanol for TRAIL, zVAD, and ivermectin, respectively.

## RNA and protein extractions

Total RNA and protein extractions were performed according to manufacturer's instructions after cells were harvested with Trizol reagent. RNA pellets were dissolved in DEPC water and protein pellets were dissolved in 1% SDS overnight at 50°C before spinning down insoluble material at 10,000 x g for 10 min.

## TUNEL assay

Terminal deoxynucleotidyl transferase Br-dUTP nick end labeling was performed on TRAIL-treated cells using a TUNEL assay kit according to manufacturer protocol. Br-dUTP incorporation was quantified by flow cytometry, analyzing the elevated peak in FL2 fluorescence on a BD Accuri Flow Cytometry System using FlowJo analysis software.

## Quantitative reverse transcription PCR (RT-qPCR)

Extracted RNA was DNAse-treated treated, primed with random nonamers, and reverse-transcribed to cDNA with Avian Myeloblastosis Virus Reverse Transcriptase according to manufacturer protocols. Genes were quantified by RT-qPCR using iTaq Universal SYBR Master Mix and primers specific to each gene of interest. RNA fold change values were calculated in reference to *18S* or *U6* ncRNAs, as indicated on figure axes. Primer sequences are listed in *Supplementary file 1B*.

## Western blotting

Protein samples were quantified by Bradford assay according to manufacturer instructions. A total of 12.5–50 µg of protein was mixed with one third volume of 4X Laemelli Sample Buffer (Bio-Rad Laboratories 1610747) and boiled at 100°C before loading into a polyacrylamide gel alongside either the PageRuler or PageRuler Plus Prestained Protein Ladder. Proteins were separated by electrophoresis and transferred to a nitrocellulose membrane, which was then cut into sections surrounding the size of the protein of interest, allowing for multiple proteins to be quantified from one gel. Membranes were then blocked with 5% non-fat dry milk in TBST (1X Tris-buffered saline with 0.2% [v/v] Tween 20) at RT for 1 hr then incubated with relevant primary antibodies diluted with 1% milk in TBST overnight at 4°C. All primary antibodies were applied at a 1:1000 dilution with the exception of the following (target, dilution): RPB1, 1:500; TUBA1A, 1:500; RPB2, 1:500; RPB3, 1:10000; LMNB1, 1:10000; GAPDH, 1:5000; and CYTC, 1:500. After three 5 min TBST washes, species-specific secondary antibodies were diluted 1:5000 with 1% milk in TBST and incubated for 1 hr at RT. Blots were then developed, after three additional 5 min TBST washes, with Clarity Western ECL Substrate for 5 min and imaged using a ChemiDoc MP Imaging System (Bio-Rad Laboratories). Each membrane section was imaged and processed separately. Band intensity was quantified using Bio-Rad Image Lab software, and relative expression changes were calculated after normalizing to an α-tubulin (TUBA1A), vinculin (VCL), or lamin-B1 (LMNB1) loading control. For all blots that appear in figures, auto-contrast was applied in Image Lab for each membrane section before the resultant image was exported for publication. When appropriate, membrane sections were stripped with 25 mM glycine in 1% SDS, pH 2 and washed two times for 10 min with TBST before being blocked and re-probed as previously described with a primary antibody targeting a protein of similar size.

## Antibody generation

Polyclonal rabbit anti-muSOX antibody was made and purified by YenZym antibodies, LLC from recombinant muSOX protein.

## Cell fractionations

For experiments performed in apoptotic HCT116 and HeLa cells, nuclear, cytoplasmic, and mitochondrial fractions were isolated using the Abcam Cell Fractionation Kit according to manufacturer's

instructions. Cytoplasmic and nuclear fractions of samples in experiments that did not involve apoptosis were separated using the REAP method (*Suzuki et al., 2010*). 1/5th of the total cell lysate was reserved and diluted to the same volume of the cell and nuclear fractions for whole cell lysate samples. Protein was extracted from 200 µL of each fraction from both methods using Trizol LS reagent and analyzed by western blot as described above.

## 4-Thiouridine- (4sU)- pulse labeling

4sU-pulse labeling was used to measure nascent transcription concurrently with mRNA decay. Fifty µM 4sU was added to cells 20 min before harvesting lysates for RNA and/or protein extraction. Labeled transcripts contained in 25 µg of total extracted RNA were biotinylated as described by *Dölken, 2013*, using 50 µg HPDP biotin. Biotinylated RNA was conjugated to Dynabeads MyOne Streptavidin C1 magnetic beads for 1 hr in the dark, then the beads were washed four times (twice at 65°C and twice at RT) with wash buffer (100 mM Tris-HCl, 10 mM EDTA, 1 M sodium chloride, and 0.1% Tween 20) before eluting RNA off of the beads twice with 5% BME in DEPC water. RNA was precipitated by adding 1/10th volume of 3M sodium acetate and 2.5 volumes of ethanol and spun down at full speed in a 4°C benchtop centrifuge. After a 75% ethanol wash, the 4sU-labeled RNA was resuspended in 20 µL DEPC water for use in in RT-qPCR.

## Reverse transcription PCR (RT-PCR)

4sU-labeled RNA from HCT116 cells treated with 100 ng/µL TRAIL, 10 µM raptinal, or their respective mock treatments was isolated as previously described. Two µL 4sU RNA from each sample was reverse transcribed into cDNA and amplified with primers targeting regions of the 18S rRNA and/or U6 snRNA using the QIAGEN OneStep RT-PCR Kit according to manufacturer instructions. Twenty-five µL of each resultant PCR product was combined with 5 µL 6X DNA loading dye and loaded onto a 1% agarose (in 1X Tris-borate EDTA) electrophoresis gel stained with SYBR Safe DNA Gel Stain alongside a DNA ladder. Gels were imaged on ChemiDoc MP Imaging System.

## 4sU-sequencing (4sU-seq)

$5 \times 10^6$ HCT116 cells were seeded onto 15 cm$^2$ plates and 24 hr later, were pre-treated with either DMSO or 20 µM zVAD, and treated with storage and dilution buffer or 100 ng/µL TRAIL for 2 hr. This process was repeated to generate two biological replicates. 50 µM 4sU was added to cells 20 min before harvesting. Cells were suspended in 2 mL Trizol and RNA extracted as previously described. Biotinylation and strepdavidin selection was performed on 200 µg of total RNA, scaling up the previously detailed protocol by 8X. 125 ng of 4sU-labeled RNA was used to synthesize rRNA-depleted sequencing libraries using KAPA-stranded RNA-Seq Kit with Ribo-Erase, HMR according to manufacturer's instructions. ERCC RNA Spike-in Mix one was added at a 1:100 dilution to each RNA sample immediately prior to library preparation normalize read counts to RNA input across samples. Libraries were submitted for analysis on a Bioanalyzer to ensure ~400 bp fragment lengths, then submitted for sequencing on a Nova-Seq 6000 with 100 bp paired-end reads at the QB3-Berkeley Genomics Sequencing Core.

Bioinformatics analysis was conducted using the UC Berkeley High Performance Computing Cluster. Paired end sequence FASTQ files were downloaded and checked for quality using FastQC. Reads were then trimmed of adaptors using Sickle/1.33. Reads were mapped to human reference genome hg19 and ERCC spike-in list was obtained using STAR genome aligner (*Dobin et al., 2013*). Differential expression upon TRAIL treatment for each gene were calculated using Cuffdiff 2 (*Trapnell et al., 2013*) on samples in the DMSO condition with their replicates, and on zVAD samples and their replicates. Differential expression values for each gene were normalized to ERCC spike-in controls. Normalized fold change values were calculated and analyzed using Microsoft Excel. Statistically significant >2 fold upregulated genes upon TRAIL treatment in the DMSO condition were input into PANTHER-GO Slim gene ontology analysis (*Mi et al., 2019*) and ChEA3 transcription factor enrichment analysis (*Keenan et al., 2019*).

## Chromatin immunoprecipitation (ChIP)

HCT116 cells nucleofected with the relevant siRNAs were seeded onto 10 cm$^2$ plates. 72 hr post-nucleofection, cells were washed with PBS, trypsinized, washed with PBS again, and fixed in 1%

formaldehyde for 2.5 min before quenching with 125 mM glycine. After an additional PBS wash, cells were lysed for 10 min at 4°C in lysis buffer (50 mM HEPES pH 7.9, 140 mM NaCl, 1 mM EDTA, 10% [v/v] glycerol, 0.5% NP40, 0.25% Triton X-100) then washed with wash buffer (10 mM Tris Cl pH 8.1, 100 mM NaCl, 1 mM EDTA pH 8.0) at 4°C for 10 min. Cells were then suspended in 1 mL shearing buffer (50 mM Tris Cl pH 7.5, 10 mM EDTA, 0.1% [v/v] SDS) and sonicated in a Covaris S220 sonicator (Covaris, Inc) with the following parameters: peak power, 140.0; duty factor, 5.0; cycle/burst, 200; and duration, 300 s. Insoluble material in the shearing buffer was then spun down at full speed in a 4°C benchtop centrifuge to yield the chromatin supernatant. Ten µg chromatin was rotated overnight at 4°C with 2.5 µg or 4 µg ChIP-grade primary antibodies targeting RPB1 and TBP, respectively, in 500 µL dilution buffer (1.1% [v/v] Triton-X-100, 1.2 mM EDTA, 6.7 mM Tris-HCl pH 8.0, 167 mM NaCl). Five µL of each IP was reserved as an input sample before antibody was added. 20 µL of either Dynabeads Protein G or a 1:1 mixture Protein A and Protein G beads, for anti-mouse and anti-rabbit antibodies, respectively, were added to each reaction and rotated at 4°C for at least 2 hr. The beads were then sequentially washed with low salt immune complex wash buffer (0.1% [v/v] SDS, 1% [v/v] Triton-X-100, 2 mM EDTA, 20 mM Tris-HCl pH 8.0, 150 mM NaCl), high low-salt immune complex wash buffer (0.1% [v/v] SDS, 1% [v/v] Triton-X-100, 2 mM EDTA, 20 mM Tris-HCl pH 8, 500 mM NaCl), LiCl immune complex buffer (0.25 M LiCl, 1% [v/v] NP40, 1% [v/v] deoxycholic acid, 1 mM EDTA, 10 mM Tris-HCl pH 8.0), and TE buffer (10 mM Tris-HCl pH 8.0, 1 mM EDTA). All washes were 5 min in duration and performed at 4°C. Beads and input samples were then suspended in 100 µL elution buffer (150 mM NaCl, 50 µg/ml proteinase K) and incubated at 55°C for 2 hr then at 65°C for 12 hr in a thermal cycler. DNA fragments were purified with an oligonucleotide clean and concentrator kit and % input values were quantified by RT-qPCR as previously described using primers complementary to the locus of interest.

## Data visualization

Bar graphs were created using GraphPad Prism eight software and the graphical abstract was created using the Bio-Render online platform.

## Quantification and statistical analysis

Biological replicates were defined as experiments performed separately on biologically distinct (i.e. from cells cultured at different times in different flasks or wells) samples representing identical conditions and/or time points. See figures and figure legends for the number of biological replicates performed for each experiment and *Supplementary file 1A* for statistical tests. Criteria for the inclusion of data was based on the performance of positive and negative controls within each experiment. No outliers were eliminated in this study. One-sample t-tests were performed on control and experimental groups for which mean fold change values were calculated, comparing these values to the null hypothesis of 1. Student's T tests (corrected for multiple comparisons with the Holm-Sidak method when appropriate) were also performed comparing means between control and experimental groups, signified by brackets spanning the two groups being compared. All statistical analyses were performed using GraphPad Prism eight unless otherwise noted.

## Acknowledgements

This work was supported by NIH grant R01CA136367 to BG and the HHMI Gilliam Fellowship for Advanced Study to CD. BG is an investigator of the Howard Hughes Medical Institute. We thank the UC Berkeley Cell Culture Facility for providing the cell lines used in this study, in addition to the UC Berkeley DNA Sequencing Facility and all members of the Glaunsinger and Coscoy labs for providing valuable feedback.

## Additional information

### Funding

| Funder | Grant reference number | Author |
| --- | --- | --- |
| National Institutes of Health | R01CA136367 | Britt A Glaunsinger |

| Howard Hughes Medical Institute | | Britt A Glaunsinger |
| Howard Hughes Medical Institute | Gilliam Fellowship | Christopher Duncan-Lewis |

The funders had no role in study design, data collection and interpretation, or the decision to submit the work for publication.

### Author contributions
Christopher Duncan-Lewis, Conceptualization, Formal analysis, Funding acquisition, Validation, Investigation, Methodology, Writing - original draft; Ella Hartenian, Resources, Investigation, Writing - review and editing; Valeria King, Formal analysis, Writing - review and editing; Britt A Glaunsinger, Conceptualization, Supervision, Writing - review and editing

### Author ORCIDs
Christopher Duncan-Lewis (ID) https://orcid.org/0000-0001-7775-5856
Britt A Glaunsinger (ID) https://orcid.org/0000-0003-0479-9377

### Decision letter and Author response
Decision letter https://doi.org/10.7554/eLife.58342.sa1
Author response https://doi.org/10.7554/eLife.58342.sa2

## Additional files

### Supplementary files
• Supplementary file 1. Statistical tests and PCR primers. (A) p values calculated by statistical tests employed in this study. (B) RT-(q)PCR primer sequences

• Supplementary file 2. GO enrichment anaylsis for subset of genes represented in the transcripts downregulated >2 fold upon TRAIL treatment (with 1 hr DMSO pre-treatment). No statistically significant (FDR < 0.05) enrichments were identified.

• Transparent reporting form

### Data availability
All data generated or analysed during this study are included in the manuscript and supporting files.

The following dataset was generated:

| Author(s) | Year | Dataset title | Dataset URL | Database and Identifier |
|---|---|---|---|---|
| Duncan-Lewis C, Hartenian E, King V, Glaunsinger B | 2020 | Cytoplasmic mRNA decay represses RNAPII transcription during early apoptosis | https://www.ncbi.nlm.nih.gov/geo/query/acc.cgi?acc=GSE163923 | NCBI Gene Expression Omnibus, GSE163923 |

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

# Appendix 1

**Appendix 1—key resources table**

| Reagent type (species) or resource | Designation | Source or reference | Identifiers | Additional information |
|---|---|---|---|---|
| Antibody | Anti-RNA polymerase II CTD repeat YSPTSPS (mouse monoclonal) | Abcam | Cat#ab817 | WB: (1:500) ChIP: (1:200) |
| Antibody | Anti-RNA polymerase II CTD repeat YSPTSPS (phospho S2) (rabbit polyclonal) | Abcam | Cat#ab5095: RRID:AB_304749 | WB: (1:1000) |
| Antibody | Anti-RNA polymerase II CTD repeat YSPTSPS (phospho S5) (rabbit polyclonal) | Abcam | Cat#ab5131; RRID:AB_449369 | WB: (1:1000) |
| Antibody | Anti-alpha Tubulin (mouse monoclonal) | Abcam | Cat#ab7291; RRID:AB_2241126 | WB: (1:500) |
| Antibody | Anti-POLR2B (RPB2) (mouse monoclonal) | Santa Cruz Biotechnology | Cat#sc-166803; RRID: AB_2167499 | WB: (1:500) |
| Antibody | Anti-RPB3 (rabbit monoclonal) | Abcam | Cat#ab182150 | WB: (1:10000) |
| Antibody | Anti-POLR2D (RPB4) (rabbit polyclonal) | ThermoFisher Scientific | Cat#PA5-35954; RRID: AB_2553264 | WB: (1:1000) |
| Antibody | Anti-Vinculin (rabbit polyclonal) | Abcam | Cat#ab91459; RRID:AB_2050446 | WB: (1:1000) |
| Antibody | Anti-PNPT1 (rabbit polyclonal) | Abcam | Cat#ab96176; RRID:AB_10680559 | WB: (1:1000) |
| Antibody | Anti-RRP41 (EXOSC4) (rabbit polyclonal) | Abcam | Cat#ab137250 | WB: (1:1000) |
| Antibody | Anti-DIS3L2 (rabbit polyclonal) | Novus Biologicals | Cat#NBP184740; RRID: AB_11038956 | WB: (1:1000) |
| Antibody | Anti-Lamin B1 (rabbit monoclonal) | Abcam | Cat#ab133741; RRID:AB_2616597 | WB: (1:10000) |
| Antibody | Anti-GAPDH (mouse monoclonal) | Abcam | Cat#ab8245; RRID:AB_2107448 | WB: (1:5000) |
| Antibody | Anti-Caspase-8 (rabbit monoclonal) | Cell Signaling Technology | Cat#4790; RRID:AB_10545768 | WB: (1:1000) |
| Antibody | Anti-Caspase-3 (rabbit polyclonal) | Cell Signaling Technology | Cat#9662; RRID:AB_331439 | WB: (1:1000) |
| Antibody | Anti-Bak (rabbit polyclonal) | Cell Signaling Technology | Cat#3814S; RRID:AB_2290287 | WB: (1:1000) |
| Antibody | Anti-Bax Antibody | Cell Signaling Technology | Cat#2772S; RRID:AB_10695870 | WB: (1:1000) |
| Antibody | Anti-XRN1 (rabbit polyclonal) | Bethyl Laboratories | Cat#A300-433A; RRID:AB_2219047 | WB: (1:1000) |
| Antibody | Anti-DFFB (rabbit polyclonal) | Abcam | Cat#ab69438; RRID:AB_2040661 | WB: (1:1000) |
| Antibody | Anti-PABP1 (rabbit polyclonal) | Cell Signaling Technology | Cat#4992; RRID:AB_10693595 | WB: (1:1000) |
| Antibody | Anti-C23 (NCL) (mouse monoclonal) | Santa Cruz Biotechnology | Cat#sc-8031; RRID:AB_672071 | WB: (1:1000) |
| Antibody | Anti-PARP (rabbit polyclonal) | Cell Signaling Technology | Cat#9542; RRID:AB_2160739 | WB: (1:1000) |
| Antibody | Anti-BID (mouse monoclonal) | Santa Cruz Biotechnology | Cat#sc-56025; RRID:AB_781628 | WB: (1:1000) |

*Continued on next page*

*Appendix 1—key resources table continued*

| Reagent type (species) or resource | Designation | Source or reference | Identifiers | Additional information |
|---|---|---|---|---|
| Antibody | Anti-TATA binding protein TBP (mouse monoclonal) | Abcam | Cat#ab51841; RRID:AB_945758 | WB: (1:1000) ChIP: (1:125) |
| Antibody | Anti-Cytochrome c (CYTC) (rabbit monoclonal) | Cell Signaling Technology | Cat#11940: RRID:AB_2637071 | WB: (1:500) |
| Antibody | Anti-VDAC1/Porin (rabbit polyclonal) | Abcam | Cat#ab15895; RRID:AB_2214787 | WB: (1:1000) |
| Antibody | Anti-muSOX (rabbit polyclonal) | This paper | N/A | WB: (1:1000) |
| Other | TrizolTM Reagent | ThermoFisher Scientific | Cat#15596026 | |
| Other | TrizolTM LS Reagent | ThermoFisher Scientific | Cat#10296028 | |
| Peptide, recombinant protein | TURBO DNase | ThermoFisher Scientific | Cat#AM2238 | |
| Peptide, recombinant protein | Avian Myeloblastosis Virus Reverse Transcriptase | Promega Corporation | Cat#M5108 | |
| Other | iTaq Universal SYBR Master Mix | Bio-Rad Laboratories | Cat#1725122 | |
| Other | Dynabeads Protein G | ThermoFisher Scientific | Cat#10003D | |
| Other | Dynabeads Protein A | ThermoFisher Scientific | Cat#10002D | |
| Other | Dynabeads MyOne Streptavidin C1 | ThermoFisher Scientific | Cat# | |
| Peptide, recombinant protein | EZ-link HPDP-biotin | ThermoFisher Scientific | Cat#21341 | |
| Peptide, recombinant protein | SuperKillerTRAIL | Enzo Life Sciences | Cat# ALX-201-115-3010 | |
| Other | KillerTRAIL Storage and Dilution Buffer | Enzo Life Sciences | Cat# ALX-505–005 R500 | |
| Chemical compound, drug | Caspase Inhibitor Z-VAD-FMK | Promega Corporation | Cat#G7231 | |
| Chemical compound, drug | Ivermectin | Millipore Sigma | Cat#I8898 | |
| Chemical compound, drug | Raptinal | Millipore Sigma | Cat#SML1745 | |
| Other | Dulbecco's Modified Eagle Medium | ThermoFisher Scientific | Cat#12800082 | |
| Other | McCoy's (modified) 5A medium | ThermoFisher Scientific | Cat#16600082 | |
| Other | Fetal Bovine Serum | VWR | Cat#89510–186 | |
| Other | Trypsin-EDTA (0.05%), phenol red | ThermoFisher Scientific | Cat# 25300120 | |
| Other | PageRuler Prestained Protein Ladder | ThermoFisher Scientific | Cat#26616 | |
| Other | PageRuler Plus Prestained Protein Ladder | ThermoFisher Scientific | Cat#26620 | |

*Continued on next page*

*Appendix 1—key resources table continued*

| Reagent type (species) or resource | Designation | Source or reference | Identifiers | Additional information |
|---|---|---|---|---|
| Other | Quick-Load Purple 1 kb Plus DNA Ladder | New England BioLabs | Cat#N0550S | |
| Other | Clarity Western ECL Substrate | Bio-Rad Laboratories | Cat#1705061 | |
| Other | 4x Laemmli Sample Buffer | Bio-Rad Laboratories | Cat#1610747 | |
| Other | Gel Loading Dye, Purple (6X) | New England BioLabs | B7025S | no SDS |
| Commercial assay, kit | TUNEL Assay Kit - BrdU-Red | Abcam | Cat#ab66110 | |
| Commercial assay, kit | OneStep RT-PCR Kit | QIAGEN | Cat#210210 | |
| Commercial assay, kit | Cell Fractionation Kit | Abcam | Cat#ab109719 | |
| Commercial assay, kit | Bio-Rad Protein Assay Kit II | Bio-Rad Laboratories | Cat#5000002 | |
| Commercial assay, kit | Oligo Clean and Concentrator Kit | Zymo Research | Cat#D4060 | |
| Commercial assay, kit | In-Fusion HD Cloning Kit | Takara Bio USA | Cat#639650 | |
| Commercial assay, kit | Lenti-X Tet-On 3G Inducible Expression System | Takara Bio USA | Cat#631187 | |
| Commercial assay, kit | Neon Transfection System | ThermoFisher Scientific | Cat#MPK5000 | |
| Commercial assay, kit | KAPA Stranded RNA-Seq Kit with RiboErase (HMR) | Roche | Cat#KK8484 | |
| Sequence-based reagent | ERCC RNA Spike-In Mix | ThermoFisher Scientific | Cat#4456740 | |
| Cell line (*Homo sapiens*) | HCT116 cells | ATCC | Cat#CCL-247; RRID: CVCL_0291 | |
| Cell line (*Homo sapiens*) | 293T/17 cells | ATCC | Cat#CRL-11268; RRID: CVCL_1926 | |
| Cell line (*Homo sapiens*) | HeLa Cells | ATCC | Cat#CCL-2; RRID:CVCL_0030 | |
| Sequence-based reagent | muSOX F | This paper | TCCCGTATACACCGG TATGTGGAGCCACCCC | |
| Sequence-based reagent | muSOX R | This paper | ATCCGCCGGCACCGG TTTAGGGGGTTATGGG | |
| Sequence-based reagent | ON-TARGETplus Non-targeting Control Pool | Horizon Discovery Group | Cat#D-001810–10 | |
| Sequence-based reagent | SMARTpool: ON-TARGETplus DIS3L2 siRNA | Horizon Discovery Group | Cat#L-018715–01 | |
| Sequence-based reagent | SMARTpool: ON-TARGETplus Human EXOSC4 siRNA | Horizon Discovery Group | Cat#L-013760–00 | |
| Sequence-based reagent | SMARTpool: ON-TARGETplus Human PNPT1 siRNA | Horizon Discovery Group | Cat#L-019454–01 | |

*Continued on next page*

*Appendix 1—key resources table continued*

| Reagent type (species) or resource | Designation | Source or reference | Identifiers | Additional information |
|---|---|---|---|---|
| Sequence-based reagent | SMARTpool: ON-TARGETplus XRN1 siRNA | Horizon Discovery Group | Cat#L-013754–01 | |
| Sequence-based reagent | SMARTpool: ON-TARGETplus CASP3 siRNA | Horizon Discovery Group | Cat#L-004307–00 | |
| Sequence-based reagent | SMARTpool: ON-TARGETplus CASP8 siRNA | Horizon Discovery Group | Cat#L-003466–00 | |
| Sequence-based reagent | ON-TARGETplus DFFB siRNA SMARTpool | Horizon Discovery Group | Cat#L-004425–00 | |
| Sequence-based reagent | SMARTpool: ON-TARGETplus Human BAX siRNA | Horizon Discovery Group | Cat# L-003308–01 | |
| Sequence-based reagent | SMARTpool: ON-TARGETplus Human BAK1 siRNA | Horizon Discovery Group | Cat# L-003305–00 | |
| Sequence-based reagent | See *Supplementary file 1* for RT-(q) PCR primers | | | |
| Software, algorithm | Prism 8 | GraphPad | RRID:SCR_002798 | https://www.graphpad.com/scientific-software/prism/ |
| Software, algorithm | FlowJo | BD | RRID:SCR_008520 | https://www.flowjo.com/solutions/flowjo |
| Software, algorithm | Image Lab Software | Bio-Rad Laboratories | Cat#1709690; RRID:SCR_014210 | |
| Software, algorithm | FastQC | Babraham Bioinformatics | RRID:SCR_014583 | http://www.bioinformatics.babraham.ac.uk/projects/fastqc |
| Software, algorithm | Sickle version 1.33 | N/A | RRID:SCR_006800 | https://github.com/najoshi/sickle |
| Software, algorithm | STAR | *Dobin et al., 2013* | RRID:SCR_004463 | https://doi.org/10.1093/bioinformatics/bts635 |
| Software, algorithm | Cuffdiff 2 | *Trapnell et al., 2013* | RRID:SCR_001647 | https://doi.org/10.1038/nbt.2450 |
| Software, algorithm | PANTHER GO-slim | *Mi et al., 2019* | RRID:SCR_002811 | http://geneontology.org/ |
| Software, algorithm | ChEA3 | *Keenan et al., 2019* | N/A | https://maayanlab.cloud/chea3/ |

