## [Decision Letter]

**Acceptance summary:**

This manuscript by Glaunsinger and colleagues examines the link between cytoplasmic RNA decay and transcriptional repression in mammalian cells undergoing apoptosis. The authors implicate mRNA decay conducted by cytoplasmic exonucleases early in apoptosis in repression of RNA PolII transcription. This response requires mitochondrial outer membrane permeabilization but not caspase activity, likely due in part to the release of the mitochondrial exonuclease PNPT1 into the cytosol.

**Decision letter after peer review:**

Thank you for submitting your article "Cytoplasmic mRNA decay represses RNA polymerase II transcription during early apoptosis" for consideration by *eLife*. Your article has been reviewed by 4 peer reviewers, and the evaluation has been overseen by Michael Green as Reviewing Editor and James Manley as the Senior Editor. The following individual involved in review of your submission has agreed to reveal their identity: Judy Lieberman.

The reviewers have discussed the reviews with one another and the Reviewing Editor has drafted this decision to help you prepare a revised submission.

Summary:

This manuscript by Glaunsinger and colleagues examines the link between cytoplasmic RNA decay and transcriptional repression in mammalian cells undergoing apoptosis. Numerous studies have found that these two processes are linked, either involving increased transcriptional activity to buffer the effects of enhanced decay or, as shown here, reinforcement of the effects of decay by coupled transcriptional repression. For that reason, further insight into how specific cellular conditions can alter the relationship between transcription and decay are important. Here, the authors implicate mRNA decay conducted by cytoplasmic exonucleases early in apoptosis in repression of RNA PolII transcription. This response requires mitochondrial outer membrane permeabilization but not caspase activity, likely due in part to the release of the mitochondrial exonuclease PNPT1 into the cytosol. Using metabolic labeling of nascent transcripts and ChIP assays of RNAPII and TBP recruitment to promoters, the authors provide evidence that transcriptional inhibition is effected at the level of pre-initiation complex recruitment. Finally, the authors show that a similar process, dependent on importin activity and involving PABPC1 relocalization to nuclei, occurs in response to herpesvirus endonuclease expression. Overall, this is an interesting study that will be of interest to a broad audience.

Essential revisions:

1. Figure 1A shows the general set-up of the death pathway, and Figure 1B repeats older data with loss of proCasp8 and the processing of Casp3. Only the cleaved form or CASP3 is actually detected, so wording, "showed activation of CASP8 and caspase 3 (CASP3) by 1.5 hr" needs to include reference to TRAIL inducing rapid apoptosis in HCT116 cells – see Figure 8 in Kim et al., 2000 as well as the Thomas et al., 2015, referenced. The data in Figure 1A and B reinforces an established property of HCT116 cells, so may be more appropriate to place in the Supplemental Information.

2. More problematically, the 4sU labeling scheme is not described clearly anywhere in the manuscript. The time of the 20 min pulse should have been more specifically described than "the last 20 min of TRAIL treatment", even in the Methods section, which is disconcerting. This is a powerful method that promises to reveal both patterns of mRNA decay but also altered RNA pol II incorporation. 4sU should be added for 20 min pulse steps from the start of TRAIL treatment through 1.5 or 2 h, with pulse and chase periods. The results of incorporation and chase should be shown for the assay genes to give a more complete picture right at the start of this story, and all in Figure 1.

3. Further, the authors' point, "It is caspase 8 (CASP8)-induced MOMP that stimulates mRNA decay in response to an apoptosis-inducing ligand (Figure 1A), partly by releasing the mitochondrial 3'-5' RNA exonuclease PNPT1 into the cytoplasm" seems to go in a novel direction. While the MOMP has clearly been shown to be important, and forms during intrinsic as well as extrinsic apoptosis in HCT116 cells, the point made in the introduction of Liu et al., 2018 is salient here, "mRNA decay occurs early after apoptosis triggered by diverse classical apoptotic stimuli (cytotoxic attack, death receptor ligation, staurosporine [STS], etoposide, tunicamycin, and thapsigargin), before membrane lipid scrambling, DNA fragmentation, and inactivation of translation initiation factors." CASP8 would be expected to be dispensable in most of these settings, although it could be activated outside of its direct role in extrinsic apoptosis as observed in other settings. With TRAIL signal transduction, CASP8 may directly activate CASP3 as well as to cleave BID and activate BAX and BAK. Emphasis on the "CASP8-induced MOMP" requires greater focus is needed here on cells deficient in steps beyond the MOMP to show that this is indeed the case, just as pursued in Thomas et al., 2015, and pointed out in Liu et al., 2018 "Apoptosis-related mRNA decay depends on MOMP-it does not occur when MOMP is blocked in BCL2-overexpressing or BAX^-/-^BAK^-/-^ cells."

4. (ln 155) siRNA knock down of CASP8 and CASP3 revealed "transcriptional repression (Figure 2D) required CASP8 but not CASP3". The authors argue that this is "in agreement with the prior study showing that MOMP-induced mRNA decay occurs before DNA fragmentation begins during extrinsic apoptosis (Thomas et al., 2015)", but that study specifically implicated the MOMP. More needs to be done here. This study demands cells that lack CASP8 as well as CASP3. The reliance on siRNA is insufficient for conclusive evidence either way. There are now not only CRISPR/Cas9 strategies that have been applied to cultured cells, but also viable mouse strains that lack CASP8 (requiring combined elimination of necroptotic machinery) and these should be used to probe the importance of this phenomenon in additional cells as well as an intact animal. Any requirement for CASP8 needs a lot more dissection, particularly with knock-out cells and animals where this protease has been eliminated without sensitizing to necroptosis (machinery that HCT116 cells apparently lack).

5. (ln 184) Despite the observations on CASP8, the conclusion, "these data suggest that neither the mRNA degradation nor the concurrent transcriptional repression observed during early apoptosis are a consequence of caspase activation" is at odds with the data shown.

6. Authors go on to hypothesize "that TRAIL-induced mRNA turnover was functionally linked to the concurrent transcriptional repression", pursuing previously identified targets, cytoplasmic 3'-RNA exonuclease DIS3L2 and the mitochondrial 3'-RNA exonuclease PNPT1", which contrasts "basal mRNA decay, which occurs predominantly from the 5' end by XRN1". This siRNA knock-down certainly reinforces the prior conclusions that 3' degradation by DIS3L2 and PNPT1 predominates in this setting.

7. Figure 3 then extends into novel territory, showing evidence that RNA pol II loading is likely compromised by TRAIL signaling at or before the formation of the preinitiation complex (PIC) of the genes assessed. The data are carefully assembled but require more precise conditions where the death pathway does not proceed beyond certain defined points (as mentioned above). The reliance on siRNA here and in Figure 4 remains a concern. This section of the manuscript promises novel and significant insights but must bring the reader to understand what step in cell death signaling drives the RNA pol II impact on initiation. The nascent 4sU pulse is appropriate and important here.

8. Figure 4 turns finally to the contributions of importans and of the viral endoribonuclease muSOX without coming to a precise synthesis of data. Complications to a simple story include the fact that mRNA degradation and RNA pol II impacts require considerably more data to provide a clear picture here. The SOX MHV68, like the homologs in Kaposi's sarcoma-associated herpesvirus and in Epstein-Barr virus, as well as the classic virion host shut-off (VHS) function encoded by in herpes simplex viruses (an analogous endoribonuclease that feeds into Xrn1-mediated 5' decay) may have an impact on RNA pol II, but this would require a bit more systematic study to be convincing.

9. The decrease in mRNA levels is measured for a few housekeeping genes only and represented as fold changes from C_t_ values normalized to 18S rRNA in reference to mock-treated cells. The graph represents mean +/-SEM, used statistic – one-sample t-test, with the hypothesis that there is no deviation from 1. Thus the representation of results does not show the variability of measurements in the reference sample (the reference sample is set to have value 1). Measurements should be presented as relative mRNA levels with appropriate statistical tests. More importantly, the RT-qPCR analysis of a few genes usually does not allow concluding that there is a global RNA-decay.

10. Transcription shut down is measured by RT-qPCR on 4sU labeled RNA, expected to represent nascent RNA. Surprisingly, the authors used the same primers for the analysis of 4sU labeled samples as used for standard RT-qPCRs. Those primers span the exon-exon junctions and are not suitable for the analysis of the nascent transcription. Apart from RT-qPCR, the Authors used RNAPII ChIP qPCRs (fig3D) using an antibody recognizing hypophosphorylated RNAPII, which normally is not engaged in transcription. Thus, such analyses are not optimal for studying the activity of RNAPII. In sum, the transcription shut down, and the rescue by 3' to 5' RNA decay nucleases is not sufficiently supported by the data. The best would be to perform a genome-wide analysis of RNA polymerase activity employing one of the broadly used techniques, for instance, GRO-seq.

11. Although the main claim of the paper is that cytoplasmic 3' exonucleases are required for apoptotic RNAPII repression, there is no explanation of why the silencing of the main cytoplasmic 5' to 3' exonuclease, Xrn1, has no effect on transcription. Moreover, all 3' exonucleases (DIS3L2, PNPT1, and the exosome subunit EXOSC4) are always silenced together and never individually. Why is it so? What if, in reality, only one nuclease is responsible for the observed effect? Importantly, there is no rescue experiment. Thus, observed effects can be attributed to off-targets of one of theses three siRNAs, especially that the mechanism of the repression remains to be elucidated.

12. A lot of attention is given to PABPC1, which upon apoptosis translocates to the nucleus (Figure 4A). Depletion of PABPC1 and PABPC4 (Figure S3A and B, this is the wrong numeration of figures probably Figure S4?) is supposed to rescue mRNA transcription, but keep reduced mRNA baseline reduced. PABPC1/4 depletion leads to a drastic reduction of mRNA levels. Thus it has a profound effect on cell physiology, which makes functional conclusion very difficult to draw, especially that they are not consistent with ivermectin treatment (see below). This part should be explored more thoughtfully or removed from the paper. At present, it adds very little to the story.

13. Importins α/β are supposed to links mRNA decay and transcription. Treatment with ivermectin, an inhibitor of α/β transport, efficiently block nucleolin localization Figure 4B and is supposed to rescue RNAPII transcription Figure 4C. Surprisingly, there is no influence on PABPC1 localization (FigS4F). Thus, on the one hand, the block of import by ivermectin rescues reduced transcription but does not influence PABPC1 relocation to the nucleus. On the other hand, depletion of PABPC1 also diminishes reduced transcription, and there is a coincidence of transcriptional repression caused by apoptosis induction and PABPC1 relocation to the nucleus. This discrepancy should be discussed.

14. Using the HCT116 cell line treated with TRAIL as a model, the Authors observed casp8 and 3 cleavages after 1.5h (Figure 1.B). They claim that casp8 stimulates mRNA decay inducing MOMP (mitochondrial outer membrane permeabilization) partly by releasing the mitochondrial 3'-5' RNA nuclease PNPT1 (two citations). Since PNPT1 activity is important for the story, the Authors should validate this aspect in their model.

15. The assays of steady-state and nascent RNA abundance that form the backbone of the paper rely on normalization to 18S (or in a few cases U6) RNA. The interpretation of these experiments relies on the assumption that levels or transcription of these ncRNAs are not affected by the cellular conditions studied, but this is not substantiated, and the rationale/validity of these controls is not discussed. The authors should provide data supporting the choice of normalization controls, such as quantification of transcripts/cell by RT-PCR or RNA-FISH.

Encouraged but optional major revisions:

1. The authors argue that RNA decay specifically represses polII transcription, but they observe reduced recruitment of TBP, which has a role in transcription by all three eukaryotic RNA polymerases. Does induction of apoptosis only affect TBP recruitment to polII promoters, or is recruitment to polI and polIII promoters also affected?

2. Figure 4: The authors tested whether importin α/β was "required for feedback between viral nuclease-driven mRNA decay and RNAPII transcription, as this would suggest that the underlying mechanisms involved in activating this pathway are conserved." I think this overstates the evidence – import is so general that it's a stretch to say that this is evidence that the underlying mechanisms are conserved.

---

## [Author Response]

Essential revisions:1. Figure 1A shows the general set-up of the death pathway, and Figure 1B repeats older data with loss of proCasp8 and the processing of Casp3. Only the cleaved form or CASP3 is actually detected, so wording, "showed activation of CASP8 and caspase 3 (CASP3) by 1.5 hr" needs to include reference to TRAIL inducing rapid apoptosis in HCT116 cells – see Figure 8 in Kim et al., 2000 as well as the Thomas et al., 2015, referenced. The data in Figure 1A and B reinforces an established property of HCT116 cells, so may be more appropriate to place in the Supplemental Information.

We have changed the wording and included the new references as suggested (Line 118). We believe that 1A and 1B provide important context to understand the data in 1C, 1D, and in subsequent figures and therefore think that these panels should remain as part of Figure 1.

2. More problematically, the 4sU labeling scheme is not described clearly anywhere in the manuscript. The time of the 20 min pulse should have been more specifically described than "the last 20 min of TRAIL treatment", even in the Methods section, which is disconcerting. This is a powerful method that promises to reveal both patterns of mRNA decay but also altered RNA pol II incorporation. 4sU should be added for 20 min pulse steps from the start of TRAIL treatment through 1.5 or 2 h, with pulse and chase periods. The results of incorporation and chase should be shown for the assay genes to give a more complete picture right at the start of this story, and all in Figure 1.

We apologize for any confusion and have now clarified our protocol, which is 4sU pulse labeling, not a 4sU pulse-chase experiment. All 4sU pulse experiments detailed in this study were performed by adding 50 mM 4sU to cell culture media 20 minutes before cells were harvested for RNA and/or protein (i.e. the last 20 min of treatment). By measuring nascent 4sU incorporation at each apoptotic timepoint, we quantified the difference in RNAPII transcriptional output that occurs in concert with corresponding levels of mRNA depletion. Chase experiments, although informative in determining the half-lives of nascent transcripts, are not required to measure transcriptional output in the context of enhanced mRNA turnover and would require extending the time of TRAIL treatment much further past the instigation of mRNA decay at 1.5-2 h, which would likely introduce more advanced features of apoptosis (such as EIF2α phosphorylation, cleavage of rRNA, and dissolution of the nuclear membrane) that could confound our analyses. We note that this 4sU pulse labeling protocol is regularly used by our group and others for transcriptional measurements (Abernathy et al., 2015; Biasini and Marques, 2020; Kenzelmann et al., 2007). The observation that during early apoptosis we detect similar reduction in the 4sU-labeled mRNA when we use intron-spanning primers to amplify the *ACTB* pre-mRNA (see new Figure 2—figure supplement 1B) as well as the observation of reduced RNAPII promoter occupancy further bolster our conclusion that transcriptional repression is occurring in addition to mRNA decay.

3. Further, the authors' point, "It is caspase 8 (CASP8)-induced MOMP that stimulates mRNA decay in response to an apoptosis-inducing ligand (Figure 1A), partly by releasing the mitochondrial 3'-5' RNA exonuclease PNPT1 into the cytoplasm" seems to go in a novel direction. While the MOMP has clearly been shown to be important, and forms during intrinsic as well as extrinsic apoptosis in HCT116 cells, the point made in the introduction of Liu et al., 2018 is salient here, "mRNA decay occurs early after apoptosis triggered by diverse classical apoptotic stimuli (cytotoxic attack, death receptor ligation, staurosporine [STS], etoposide, tunicamycin, and thapsigargin), before membrane lipid scrambling, DNA fragmentation, and inactivation of translation initiation factors." CASP8 would be expected to be dispensable in most of these settings, although it could be activated outside of its direct role in extrinsic apoptosis as observed in other settings. With TRAIL signal transduction, CASP8 may directly activate CASP3 as well as to cleave BID and activate BAX and BAK. Emphasis on the "CASP8-induced MOMP" requires greater focus is needed here on cells deficient in steps beyond the MOMP to show that this is indeed the case, just as pursued in Thomas et al., 2015, and pointed out in Liu et al., 2018 "Apoptosis-related mRNA decay depends on MOMP-it does not occur when MOMP is blocked in BCL2-overexpressing or BAX^-/-^BAK^-/-^ cells."

We fully agree that it is MOMP, whether induced by CASP8 or other stimuli, that is responsible for mRNA decay-induced transcriptional repression. Our qualifier that CASP8 was required “in response to an apoptosis-inducing ligand” was meant to distinguish CASP8-induced MOMP from that caused by intrinsic apoptosis inducers. To make this point more clearly, we have changed the wording of the referenced sentence to clarify that CASP8 is not necessarily required for MOMP (line 112). We also included an experiment in our initial submission using a small molecule, raptinal, that directly instigates MOMP in order to demonstrate that the mRNA decay and ensuing RNAPII transcriptional repression observed during intrinsic apoptosis and are not affected by the pan-caspase inhibitor zVAD (now Figure 3E-F). In order to more definitively link the transcriptional repression to MOMP-induced mRNA decay, we now include an additional set of experiments in which we show that depletion of the mitochondrial pore-forming proteins BAX and BAK inhibit TRAIL-induced mRNA decay and transcriptional repression (Figure 3C-D, Figure 3—figure supplement 1D).

4. (ln 155) siRNA knock down of CASP8 and CASP3 revealed "transcriptional repression (Figure 2D) required CASP8 but not CASP3". The authors argue that this is "in agreement with the prior study showing that MOMP-induced mRNA decay occurs before DNA fragmentation begins during extrinsic apoptosis (Thomas et al., 2015)", but that study specifically implicated the MOMP. More needs to be done here. This study demands cells that lack CASP8 as well as CASP3. The reliance on siRNA is insufficient for conclusive evidence either way. There are now not only CRISPR/Cas9 strategies that have been applied to cultured cells, but also viable mouse strains that lack CASP8 (requiring combined elimination of necroptotic machinery) and these should be used to probe the importance of this phenomenon in additional cells as well as an intact animal. Any requirement for CASP8 needs a lot more dissection, particularly with knock-out cells and animals where this protease has been eliminated without sensitizing to necroptosis (machinery that HCT116 cells apparently lack).

To clarify, we are not arguing for a direct role for CASP8 (or any other caspase) in mRNA decay or transcription, as the pathway can be activated by inducers of MOMP that bypass CASP8. Instead, our data demonstrate that CASP8 is only involved to the extent that it sets off a signaling cascade that leads to MOMP (and by extension, mRNA degradation) when activated by a death-inducing signaling complex. The distinction between CASP8 and CASP3 is only made to differentiate the action of CASP8 (which cleaves a limited set of targets and instigates MOMP-induced mRNA degradation in response to TRAIL) from that of the mass proteolysis of CASP3.

The data supporting this assertion are: (a) CASP3 knock down does not affect mRNA decay or transcription while knock down of CASP8 rescues both in TRAIL treated cells, (b) inducing MOMP directly with a small molecule prompts mRNA decay and RNAPII transcriptional repression even in the presence of a pan-caspase inhibitor, and (c) our new data showing that depletion of mitochondrial pore-forming proteins BAX and BAK is sufficient to attenuate both mRNA decay and RNAPII transcriptional repression, even though caspases are still activated in their absence. We believe that these experiments clearly demonstrate the necessity and sufficiency of MOMP-induced mRNA decay (and not necessarily caspase activity) in reducing mRNA output in apoptotic cells.

5. (ln 184) Despite the observations on CASP8, the conclusion, "these data suggest that neither the mRNA degradation nor the concurrent transcriptional repression observed during early apoptosis are a consequence of caspase activation" is at odds with the data shown.

We hope this comment (which is linked to point 4) was clarified above. We have also amended the wording of this sentence to state that “caspases are not required to drive these phenotypes outside of their role in MOMP activation.” (Lines 239-240).”

6. Authors go on to hypothesize "that TRAIL-induced mRNA turnover was functionally linked to the concurrent transcriptional repression", pursuing previously identified targets, cytoplasmic 3'-RNA exonuclease DIS3L2 and the mitochondrial 3'-RNA exonuclease PNPT1", which contrasts "basal mRNA decay, which occurs predominantly from the 5' end by XRN1". This siRNA knock-down certainly reinforces the prior conclusions that 3' degradation by DIS3L2 and PNPT1 predominates in this setting.7. Figure 3 then extends into novel territory, showing evidence that RNA pol II loading is likely compromised by TRAIL signaling at or before the formation of the preinitiation complex (PIC) of the genes assessed. The data are carefully assembled but require more precise conditions where the death pathway does not proceed beyond certain defined points (as mentioned above). The reliance on siRNA here and in Figure 4 remains a concern. This section of the manuscript promises novel and significant insights but must bring the reader to understand what step in cell death signaling drives the RNA pol II impact on initiation. The nascent 4sU pulse is appropriate and important here.

Having demonstrated that the action of mRNA decay factors drives the RNAPII transcriptional repression upon TRAIL treatment, we focused on the effects of the decay factors on RNAPII promoter occupancy during early apoptosis. We now include additional RNAPII ChIP experiments in cells depleted of CASP8 and CASP3 that validate concordance with the decreases in 4sU incorporation under these conditions (see new Figure 4—figure supplement 1C).

Our choice to use siRNAs instead of CRISPR/Cas9 in order to deplete cellular proteins, particularly mRNA decay factors, is informed by previous observations in our group that depletion for a sustained period of time (e.g. during single cell selection of CRISPR-based knockout clones) of individual mRNA decay factors can lead to compensatory changes in the expression levels of other mRNA decay factors. We favored siRNAs since they provide more acute depletion of protein levels (<72 h before harvesting) and do not elicit a compensatory increase in the abundance non-targeted decay factors (see Figure 4A). Furthermore, all siRNAs employed in this study were Horizon Discovery ON TARGETplus siRNAs, which utilize dual strand modifications reported to greatly reduce the off-target effects characteristic of unmodified siRNAs (Jackson et al., 2006).

8. Figure 4 turns finally to the contributions of importans and of the viral endoribonuclease muSOX without coming to a precise synthesis of data. Complications to a simple story include the fact that mRNA degradation and RNA pol II impacts require considerably more data to provide a clear picture here. The SOX MHV68, like the homologs in Kaposi's sarcoma-associated herpesvirus and in Epstein-Barr virus, as well as the classic virion host shut-off (VHS) function encoded by in herpes simplex viruses (an analogous endoribonuclease that feeds into Xrn1-mediated 5' decay) may have an impact on RNA pol II, but this would require a bit more systematic study to be convincing.

We certainly agree that if this were the only piece of data linking these viral mRNA degrading nucleases to transcriptional repression, we would not be able to draw strong conclusions! However, our group has published multiple mechanistic studies elucidating the link between viral endonucleases and RNAPII transcriptional repression. Briefly, KSHV and MHV68 infections reduce transcription of several RNAPII genes, as measured by both 4sU pulse labeling and RNAPII ChIP, in a manner dependent on catalytically active SOX or muSOX endonucleases (Abernathy et al., 2015; Hartenian et al., 2020). Ectopic expression of muSOX or HSV-1 vhs are sufficient to drive RNAPII transcriptional repression (Abernathy et al., 2015; Gilbertson et al., 2018), demonstrating that viral mRNA decay by multiple herpesviruses directly cause this phenotype. The transcriptional repression observed during infection is widespread; MHV68 infection reduces RNAPII occupancy at 86% of promoters as measured by ChIP-seq (Hartenian et al., 2020). Finally, the widespread mRNA decay induced by muSOX causes relocalization of many RNA binding proteins from the cytoplasm to the nucleus in a manner dependent on degradation of the muSOX-cleaved mRNA fragments by Xrn1 (Gilbertson et al., 2018). Here, we are adding to this significant body of work by demonstrating that the nucleo-cytoplasmic shuttling of proteins induced during mRNA decay is required for subsequent repression of RNAPII transcription. This link was hypothesized in the Gilbertson et al. publication but is experimentally demonstrated here.

9. The decrease in mRNA levels is measured for a few housekeeping genes only and represented as fold changes from C_t_ values normalized to 18S rRNA in reference to mock-treated cells. The graph represents mean +/-SEM, used statistic – one-sample t-test, with the hypothesis that there is no deviation from 1. Thus the representation of results does not show the variability of measurements in the reference sample (the reference sample is set to have value 1). Measurements should be presented as relative mRNA levels with appropriate statistical tests. More importantly, the RT-qPCR analysis of a few genes usually does not allow concluding that there is a global RNA-decay.

Our primary metric of RNA decay is the change in the abundance of each transcript upon apoptosis induction, i.e. the extent of mRNA depletion from baseline levels. Similarly, we sought to measure the change in 4sU incorporation upon apoptosis induction compared to baseline as a proxy for transcriptional repression. This method of normalization has been used in similar studies, including a recent publication in *eLife* from our group (Gilbertson et al., 2018) and the paper that elucidated the role of PNPT1 in apoptotic mRNA decay (Liu et al., 2018). In addition to one sample t tests performed to determine if any given fold change between untreated and induced apoptotic cells differed from a hypothetical fold change of one, we also used multiple t tests (corrected for multiple comparisons) to compare fold changes upon apoptosis induction in the backgrounds of cells treated with different compounds or siRNAs (e.g. comparing the pairwise differences in 4sU incorporation between untreated and TRAIL-treated cells both in the presence and absence of ivermectin). These tests are indicated with brackets between the fold changes being compared and are noted as such in the figure legends. We believe that these are accurate statistical methods that suit the data but are happy to perform any additional specific statistical tests deemed more appropriate.

In response to the comment regarding our measuring only select housekeeping genes as representative examples, we performed 4sU-sequencing to determine the extent to which mRNA synthesis is reduced during early apoptosis (new Figure 2C-F, Figure 2—figure supplement 1B-D). These data reveal that the concurrent transcriptional repression during early apoptosis is indeed global, impacting 71.2% of human genome-aligned transcripts detected.

10. Transcription shut down is measured by RT-qPCR on 4sU labeled RNA, expected to represent nascent RNA. Surprisingly, the authors used the same primers for the analysis of 4sU labeled samples as used for standard RT-qPCRs. Those primers span the exon-exon junctions and are not suitable for the analysis of the nascent transcription. Apart from RT-qPCR, the Authors used RNAPII ChIP qPCRs (fig3D) using an antibody recognizing hypophosphorylated RNAPII, which normally is not engaged in transcription. Thus, such analyses are not optimal for studying the activity of RNAPII. In sum, the transcription shut down, and the rescue by 3' to 5' RNA decay nucleases is not sufficiently supported by the data. The best would be to perform a genome-wide analysis of RNA polymerase activity employing one of the broadly used techniques, for instance, GRO-seq.

Thank you for these suggestions. As described above, we have added a 4sU-seq experiment (new Figure 2C-F, Figure 2—figure supplement 1CD) comparing the change in 4sU labelling of nascent RNA upon TRAIL treatment both in the presence and absence of the pan-caspase inhibitor zVAD in order to better assess the breadth of transcriptional repression.

While our exonic primers did amplify RNA synthesized during the short 4sU pulse, we agree that showing similar results with intronic primers would bolster confidence that the 4sU-labeled RNA represents nascent transcripts. We therefore designed intronic primers for the ACTB transcript and showed that RT-qPCR of the 4sU labeled RNA with these primers gave results similar to those with the exonic primers, in that ACTB mRNA decreased upon TRAIL treatment but was rescued in the presence of zVAD (new Figure 2—figure supplement 1B). We observed similar agreement between exonic and intronic qPCR primers used to quantify RNA isolated from a short duration 4sU pulse in a previous study from our group (Abernathy et al., 2015). This trend also held true for the next-generation sequencing data, which does not rely on gene-specific primers.

The RNAPII CTD is recruited to promoters in a hypophosphorylated state (Brien et al., 1993; Cheng and Sharp, 2003; Usheva et al., 1992), so the 8WG16 antibody is appropriate for measuring differences in RNAPII promoter recruitment. It should also be noted that this antibody does not exclusively bind hypophosphorylated RBP1, but rather exhibits a preference for this form (Nojima et al., 2015).

11. Although the main claim of the paper is that cytoplasmic 3' exonucleases are required for apoptotic RNAPII repression, there is no explanation of why the silencing of the main cytoplasmic 5' to 3' exonuclease, Xrn1, has no effect on transcription. Moreover, all 3' exonucleases (DIS3L2, PNPT1, and the exosome subunit EXOSC4) are always silenced together and never individually. Why is it so? What if, in reality, only one nuclease is responsible for the observed effect? Importantly, there is no rescue experiment. Thus, observed effects can be attributed to off-targets of one of these three siRNAs, especially that the mechanism of the repression remains to be elucidated.

The reason why XRN1 depletion does not impact transcription in this system is that apoptotic mRNA decay does not occur from the 5’ end. The Lieberman lab reported that 3’ (but not 5’) mRNA decay intermediates can be detected in apoptotic cells, and our results with XRN1 depletion are consistent with their observations. This point is clarified in the discussion: “XRN1-driven 5’-3’ end decay is the major pathway involved in basal mRNA decay (Łabno et al., 2016), but MOMP-induced mRNA decay is primarily driven by 3’ exonucleases such as PNPT1 and DIS3L2 (Liu et al., 2018; Thomas et al., 2015). Accordingly, co-depletion of 3’ decay factors but not XRN1 restored RNAPII promoter occupancy and transcription during early apoptosis.” (Lines 375-377)

While 5’ end decay is carried out by a single 5’-3’ exonuclease (XRN1), 3’ end decay is carried out by multiple 3’-5’ exonucleases that can function in at least partially redundant ways (Houseley and Tollervey, 2009). For this reason, in general 3’ end decay cannot be effectively stalled by individual enzyme depletions. Indeed, we did perform individual knockdowns of the 3’ mRNA decay factors and found that no individual knock down had a reproducible effect on ACTB and GAPDH total or 4sU-labeled mRNA levels (these data are now included as Figure 4—figure supplement 1A-B). Instead, only co-depletion of multiple 3’ end decay factors inhibited mRNA degradation and rescued mRNA transcription. The fact that none of the individual siRNAs we used altered ACTB and GAPDH levels argues against the rescue phenotype in the co-depletion experiments being due to off-target effects (as these would have manifested in the single knock downs as well).

12. A lot of attention is given to PABPC1, which upon apoptosis translocates to the nucleus (Figure 4A). Depletion of PABPC1 and PABPC4 (Figure S3A and B, this is the wrong numeration of figures probably Figure S4?) is supposed to rescue mRNA transcription, but keep reduced mRNA baseline reduced. PABPC1/4 depletion leads to a drastic reduction of mRNA levels. Thus it has a profound effect on cell physiology, which makes functional conclusion very difficult to draw, especially that they are not consistent with ivermectin treatment (see below). This part should be explored more thoughtfully or removed from the paper. At present, it adds very little to the story.

We agree that the PABPC1/4 depletion experiment adds little to the story given these caveats, so we have removed the data from the paper.

Importins α/β are supposed to links mRNA decay and transcription. Treatment with ivermectin, an inhibitor of α/β transport, efficiently block nucleolin localization Figure 4B and is supposed to rescue RNAPII transcription Figure 4C. Surprisingly, there is no influence on PABPC1 localization (FigS4F). Thus, on the one hand, the block of import by ivermectin rescues reduced transcription but does not influence PABPC1 relocation to the nucleus. On the other hand, depletion of PABPC1 also diminishes reduced transcription, and there is a coincidence of transcriptional repression caused by apoptosis induction and PABPC1 relocation to the nucleus. This discrepancy should be discussed.

We devoted much of our attention to PABPC1 since the protein has previously been shown to translocate into the nucleus in response to widespread mRNA turnover (Burke et al., 2019; Kumar et al., 2011; Kumar and Glaunsinger, 2010) and has been linked to the transcriptional repression elicited by expression of the muSOX exonuclease (Gilbertson et al., 2018). Although we have shown that PABPC1 nuclear influx also occurs during early apoptosis, a causative link between this relocalization and the coincidental reduction in RNAPII transcription has not been definitively established. We hypothesize that the observed RNAPII transcriptional repression occurs as a result of the cumulative import of multiple factors, not all of which are blocked from entering the nucleus by inhibiting only one route of nuclear import. Our group previously reported that at least 66 proteins in addition to PABPC1 are selectively enriched in the nucleus upon transfection with the viral endonuclease muSOX (but not with the catalytically-dead D219A mutant), 22 of which are known to be RNA-binding proteins and 45 of which shuttle in a manner dependent on the cytoplasmic mRNA exonuclease primarily responsible for clearing cleavage fragments, XRN1 (Gilbertson et al., 2018). The nuclear overexpression of PABPC1 alone only reduces RNAPII promoter recruitment when done to an extent much greater than that observed during muSOX expression or apoptosis induction, suggesting that these additional factors may be involved in regulating transcription physiological contexts. PABPC1/4 knockdown generally destabilizes mRNAs and restricts gene expression, so the loss of the link between mRNA degradation and RNAPII transcription that occurs in the absence of these factors could just as well be due to reduced expression of other key factors rather than a specific role of PABPC. As noted in point 12 above, we removed the PABPC1/4 depletion experiment from the paper. Future studies in which changes in the nuclear and cytoplasmic proteome upon apoptosis induction in the presence and absence of ivermectin will likely provide insight into which additional protein or proteins may play a role in connecting cytoplasmic mRNA turnover to RNAPII transcription.

14. Using the HCT116 cell line treated with TRAIL as a model, the Authors observed casp8 and 3 cleavages after 1.5h (Figure 1.B). They claim that casp8 stimulates mRNA decay inducing MOMP (mitochondrial outer membrane permeabilization) partly by releasing the mitochondrial 3'-5' RNA nuclease PNPT1 (two citations). Since PNPT1 activity is important for the story, the Authors should validate this aspect in their model.

In response to this comment, we spent several months trying to generate at PNPT1 knockout using CRISPR/Cas9 (with the goal of performing the suggested rescue experiment), however were unsuccessful at generating this line despite multiple attempts. However, we note that Liu et al. comprehensively characterized the nature of mRNA degradation that occurs after TRAIL treatment in HCT116 cells and the role of PNPT1 in this process (Liu et al., 2018). We performed a variety of experiments to validate this model of mRNA decay (TRAIL treatment with CASP3, CASP8, BAX/BAK, and EXOSC4/DIS3L2/PNPT1 knockdowns, inducing MOMP directly with raptinal), and in each case, mRNA decay was linked to and required for transcriptional repression.

15. The assays of steady-state and nascent RNA abundance that form the backbone of the paper rely on normalization to 18S (or in a few cases U6) RNA. The interpretation of these experiments relies on the assumption that levels or transcription of these ncRNAs are not affected by the cellular conditions studied, but this is not substantiated, and the rationale/validity of these controls is not discussed. The authors should provide data supporting the choice of normalization controls, such as quantification of transcripts/cell by RT-PCR or RNA-FISH.

18S rRNA levels have been previously reported to be stable during early apoptosis (Houge et al., 1995; Thomas et al., 2015), which was confirmed in our hands by near-identical 18S C_t_ values in cDNA synthesized from equal volumes of total RNA extracted from the same number of apoptotic and non-apoptotic cells. 18S C_t_ values were also stable in 4sU-labeled RNA, but since this is the first time to our knowledge that nascent transcription during early apoptosis has been measured in such a way, we have added RT-PCR agarose gels to illustrate this fact (Figure 1—figure supplement 1B). In contrast, the levels of 18S 4sU RNA seemed to decrease upon 4 hr raptinal treatment, perhaps due to the onset of 18S rRNA cleavage during later stages of apoptosis (Lafarga et al., 1997), so we instead normalized to the relatively stable U6 transcript. RT-PCR illustrating these trends have also been added (Figure 3—figure supplement 1F).

Encouraged but optional major revisions:1. The authors argue that RNA decay specifically represses polII transcription, but they observe reduced recruitment of TBP, which has a role in transcription by all three eukaryotic RNA polymerases. Does induction of apoptosis only affect TBP recruitment to polII promoters, or is recruitment to polI and polIII promoters also affected?

We believe that the defect in TBP recruitment occurs upstream of any regulation of TBP itself, perhaps at the level of chromatin availability, given that there is not a decrease in RNAPI and III transcripts known to have TBP at their promoter (such as 18S, 7SK, and U6).

2. Figure 4: The authors tested whether importin α/β was "required for feedback between viral nuclease-driven mRNA decay and RNAPII transcription, as this would suggest that the underlying mechanisms involved in activating this pathway are conserved." I think this overstates the evidence – import is so general that it's a stretch to say that this is evidence that the underlying mechanisms are conserved.

We have amended this sentence to end “may be conserved.” (Line 338-339)

References:

Abernathy, E., Gilbertson, S., Alla, R., Glaunsinger, B., 2015. Viral nucleases induce an mRNA degradation-transcription feedback loop in mammalian cells. Cell Host Microbe 18, 243–253.

Biasini, A., Marques, A.C., 2020. A Protocol for Transcriptome-Wide Inference of RNA Metabolic Rates in Mouse Embryonic Stem Cells. Front. Cell Dev. Biol. 8, 1–11.

Brien, T.O., Hardlnu, S., Greenleaft, A., Lls, J.T., 1993. Phosphorylation of RNA polymerase lie-terminal elongation 75–77.

Burke, J.M., Moon, S.L., Matheny, T., Parker, R., 2019. RNase L Reprograms Translation by Widespread mRNA Turnover Escaped by Antiviral mRNAs. Mol. Cell 75, 1203-1217.e5.

Cheng, C., Sharp, P.A., 2003. RNA Polymerase II Accumulation in the Promoter-Proximal Region of the Dihydrofolate Reductase and γ-Actin Genes. Mol. Cell. Biol. 23, 1961–1967.

Gilbertson, S., Federspiel, J.D., Hartenian, E., Cristea, I.M., Glaunsinger, B., 2018. Changes in mRNA abundance drive shuttling of RNA binding proteins, linking cytoplasmic RNA degradation to transcription. *eLife* 7, 1–26.

Hartenian, E., Gilbertson, S., Federspiel, J.D., Cristea, I.M., Glaunsinger, B.A., 2020. RNA decay during gammaherpesvirus infection reduces RNA polymerase II occupancy of host promoters but spares viral promoters. PLoS Pathog. 16, e1008269–e1008269.

Houge, G., Robaye, B., Eikhom, T.S., Golstein, J., Mellgren, G., Gjertsen, B.T., Lanotte, M., Døskeland, S.O., 1995. Fine mapping of 28S rRNA sites specifically cleaved in cells undergoing apoptosis. Mol. Cell. Biol. 15, 2051–2062.

Houseley, J., Tollervey, D., 2009. The Many Pathways of RNA Degradation. Cell 136, 763–776.

Jackson, A.L., Burchard, J., Leake, D., Reynolds, A., Schelter, J., Guo, J., Johnson, J.M., Lim, L., Karpilow, J., Nichols, K., Marshall, W., Khvorova, A., Linsley, P.S., 2006. Position-specific chemical modification of siRNAs reduces “off-target” transcript silencing. Rna 12, 1197–1205.

Kenzelmann, M., Maertens, S., Hergenhahn, M., Kueffer, S., Hotz-Wagenblatt, A., Li, L., Wang, S., Ittrich, C., Lemberger, T., Arribas, R., Jonnakuty, S., Hollstein, M.C., Schmid, W., Gretz, N., Gröne, H.J., Schütz, G., 2007. Microarray analysis of newly synthesized RNA in cells and animals. Proc. Natl. Acad. Sci. U. S. A. 104, 6164–6169.

Kumar, G.R., Glaunsinger, B.A., 2010. Nuclear Import of Cytoplasmic Poly(A) Binding Protein Restricts Gene Expression via Hyperadenylation and Nuclear Retention of mRNA. Mol. Cell. Biol. 30, 4996–5008.

Kumar, G.R., Shum, L., Glaunsinger, B.A., 2011. Importin -Mediated Nuclear Import of Cytoplasmic Poly(A) Binding Protein Occurs as a Direct Consequence of Cytoplasmic mRNA Depletion. Mol. Cell. Biol. 31, 3113–3125.

Łabno, A., Tomecki, R., Dziembowski, A., 2016. Cytoplasmic RNA decay pathways – Enzymes and mechanisms. Biochim. Biophys. Acta – Mol. Cell Res. 1863, 3125–3147.

Lafarga, M., Lerga, A., Andres, M.A., Polanco, J.I., Calle, E., Berciano, M.T., 1997. Apoptosis induced by methylazoxymethanol in developing rat cerebellum: Organization of the cell nucleus and its relationship to DNA and rRNA degradation. Cell Tissue Res. 289, 25–38.

Liu, X., Fu, R., Pan, Y., Meza-sosa, K.F., Zhang, Z., Liu, X., Fu, R., Pan, Y., Meza-sosa, K.F., Zhang, Z., Lieberman, J., 2018. PNPT1 Release from Mitochondria during Apoptosis Triggers Decay of Poly(A) RNAs. Cell 1–15.

Nojima, T., Gomes, T., Grosso, A.R.F., Kimura, H., Dye, M.J., Dhir, S., Carmo-Fonseca, M., Proudfoot, N.J., 2015. Mammalian NET-seq reveals genome-wide nascent transcription coupled to RNA processing. Cell 161, 526–540.

Thomas, M.P., Liu, X., Whangbo, J., McCrossan, G., Sanborn, K.B., Basar, E., Walch, M., Lieberman, J., 2015. Apoptosis Triggers Specific, Rapid, and Global mRNA Decay with 3’ Uridylated Intermediates Degraded by DIS3L2. Cell Rep. 11, 1079–1089.

Usheva, A., Maldonado, E., Goldring, A., Lu, H., Houbavi, C., Reinberg, D., Aloni, Y., 1992. Specific interaction between the nonphosphorylated form of RNA polymerase II and the TATA-binding protein. Cell 69, 871–881.